

# Impact of physical parameterizations and initial conditions on simulated atmospheric transport and CO₂ mole fractions in the US Midwest

Liza I. Díaz-Isaac [1,*], Thomas Lauvaux [1], Kenneth J. Davis [1]

[1]Department of Meteorology and Atmospheric Science, The Pennsylvania State University, University Park, 16803, USA
    [*] Now at Scripps Institution of Oceanography, University of California, San Diego, 92093, USA

*Correspondence to*: Liza I. Díaz-Isaac (lzd120@psu.edu)

**Abstract.** Atmospheric transport model errors are one of the main contributors to the uncertainty affecting $CO_2$ inverse flux estimates. In this study, we determine the leading causes of transport errors over the US Upper Midwest with a large set of

simulations generated with the Weather Research and Forecasting (WRF) mesoscale model. The various WRF simulations are performed using different meteorological driver datasets and physical parameterizations including planetary boundary layer (PBL) schemes, land surface models (LSMs), cumulus parameterizations and microphysics parameterizations. All the different model configurations were coupled to $CO_2$ fluxes and lateral boundary conditions from the CarbonTracker inversion system to simulate atmospheric $CO_2$ mole fractions. PBL height, wind speed, wind direction, and atmospheric $CO_2$

mole fractions are compared to observations during a month of the summer of 2008, and statistical analyses were performed to evaluate the impact of both physics parameterizations and meteorological datasets on these variables. All of the physical parameterizations and the meteorological initial and boundary conditions contribute 3 to 4 ppm to the model-to-model variability in daytime PBL $CO_2$ except for the microphysics parameterization which has a smaller contribution. PBL height varies across ensemble members by 300 to 400 m, and is this variability is controlled by the same physics parameterizations.

Daily PBL $CO_2$ mole fraction errors are correlated with errors in the PBL height. We show that specific model configurations systematically overestimate or underestimate the PBL height averaged across the region with biases closely correlated with the choice of LSM, PBL scheme, and CP. Domain average PBL wind speed is overestimated in nearly every model configuration. Both PBLH and PBL wind speed biases show coherent spatial variations across the Midwest, with PBLH overestimated averaged across configurations by 300-400 m in the west, and PBL winds overestimated by about 1

m/s on average in the east. We find model configurations with lower biases averaged across the domain, but no single configuration is optimal across the entire region and for all meteorological variables. We conclude that model ensembles that include multiple physics parameterizations and meteorological initial conditions are likely to be necessary to encompass the atmospheric conditions most important to the transport of $CO_2$ in the PBL, but that construction of such an ensemble will be challenging due to ensemble biases that vary across the region.



## 1 Introduction

The increase in atmospheric carbon dioxide ($CO_2$) mole fraction is a primary factor that is changing the radiation budget and causing significant changes in the Earth's climate (IPCC, 2013). Atmospheric mole fractions have increased primarily due to fossil fuel combustion and land use change. Not all $CO_2$ emitted remains in the atmosphere because the terrestrial biosphere

absorbs about 30% of the emissions (Le Queré et al., 2015). Terrestrial ecosystems in the temperate northern latitudes are identified as a substantial sink (Tans et al., 1990; Ciais et al., 1995; Gurney et al., 2002; Sarmiento et al., 2010; Pan et al., 2011; Le Queré et al., 2015). However, the specific magnitudes and distributions of terrestrial sources and sinks are still uncertain. Accurate and precise quantification of these fluxes is an important step towards a successful prediction of future atmospheric $CO_2$ and climate change mitigation.

One method used to estimate the terrestrial fluxes is the "top-down" or atmospheric inverse method. Atmospheric inversions use simulations of atmospheric $CO_2$ to estimate carbon fluxes (i.e., prior fluxes) by adjusting these fluxes so that simulated $CO_2$ is optimally consistent with observed $CO_2$ mole fractions (e.g., Enting, 1993; Bousquet et al., 2000; Chevallier et al., 2010).  Uncertainties in the inverse method can be caused by sparse atmospheric data (Gurney et al., 2002), uncertain prior flux estimates (Huntzinger et al., 2012), limited spatial resolution in biospheric and atmospheric models, and transport model

errors (Stephen et al., 2007; Gerbig et al., 2008; Pickett-Heaps et al., 2011; Díaz Isaac et al., 2014). Despite progress in top down methodologies, these sources of uncertainty have hindered the accuracy and precision of inverse estimates of sources and sinks from terrestrial ecosystems at continental scales (Le Quéré et al., 2015).

Current atmospheric inversion systems are limited to the optimization of surface fluxes. However, the model-data mismatches used to optimize the fluxes contain the contributions of both flux and transport errors. Therefore, the

atmospheric inversions may attribute atmospheric $CO_2$ model-data mismatches to surface fluxes. In a Bayesian framework, the atmospheric inversion assumes (1) atmospheric transport model errors are unbiased and (2) the random errors are known. Incorrectly prescribed errors (i.e., random and systematic) will be propagated into the state space by the optimization process, generating biased inverse (i.e. posterior) fluxes (Tarantola, 2005). The atmospheric inverse system will be reliable only if both the atmospheric transport random errors are quantified rigorously and the transport model is unbiased.

To date, relatively few studies have focused on atmospheric transport errors. The Atmospheric Tracer Transport Model Intercomparison Project (TransCom) has been dedicated to quantifying atmospheric transport errors and their impact on $CO_2$ fluxes through model inter-comparisons (Gurney et al., 2002; Baker et al., 2006; Stephen et al., 2007; Patra et al., 2008; Peylin et al., 2013). As inter-comparison exercises, TransCom studies were not always limited to varying atmospheric transport, but at times also varied the number of observations, the inverse methodologies, and the prior fluxes that were used.

Some of these studies have concluded that only an atmospheric transport model capable of representing synoptic and mesoscale atmospheric dynamics will be able to extract high-resolution information from atmospheric observations (Law et al., 2008; Patra et al., 2008). Following these recommendations, the spatial resolution of transport models used to simulate atmospheric $CO_2$ mole fractions has increased to capture local-scale variability in continental observations (e.g. Ahmadov et



al., 2009). Díaz Isaac et al. (2014) showed significant differences in the atmospheric $CO_2$ model-data mismatches when comparing a lower-resolution global transport model to a high-resolution regional transport model, but using identical surface fluxes, suggesting that changes in the transport model resolution could lead to large differences in inverse surface flux estimates.

A critical problem in atmospheric transport resides in the representation of vertical mixing, which significantly impacts the interpretation of near-surface $CO_2$ mole fractions and the resulting inverse $CO_2$ flux estimates (Denning et al., 1995; Stephens et al., 2007). As a result, several studies have been dedicated to the evaluation of mixed layer (ML) depth (Yi et al., 2004; Gerbig et al., 2008; Kretschmer et al., 2012). An overestimation of the ML depth by an atmospheric model, for example, will cause an overestimation of the $CO_2$ surface flux magnitude. The misrepresentation of vertical mixing by

TransCom's atmospheric models shown by Stephens et al. (2007) led Gerbig et al., (2008) to evaluate uncertainty in ML depth using a regional model and find random errors on the order of several ppm in ML $CO_2$ mole fractions in summertime over western Europe.  Sarrat et al., (2007) used an inter-comparison of five mesoscale models and identified discrepancies in the ML depth that was potentially impacting the atmospheric $CO_2$ mole fractions. These studies have attributed the differences between simulated and observed mixed ML height to flaws in planetary boundary layer (PBL) schemes and land

surface models (LSMs). The accurate representation of the ML depth, however, is a necessary but most likely insufficient step for accurate and precise simulation of $CO_2$ mole fractions in the lower troposphere.  Mixing between the ML and the rest of the atmosphere is also an important factor in the relationship between surface fluxes of $CO_2$ and ML $CO_2$ mole fractions.  It is likely that parameterizations other than the PBL and LSM will influence ML $CO_2$ mole fractions.

Inter-comparison of physical parameterization schemes using the Weather Research and Forecasting (WRF; Skamarock et

al., 2005) mesoscale model has been explored to understand the impact of physics parameterizations on the $CO_2$ mole fractions (Kretschmer et al., 2012; Yver et al., 2013; Lauvaux and Davis, 2014; Feng et al., 2016). These studies have found that parameterization choices can result in systematic errors of several ppm in atmospheric PBL $CO_2$ that can lead to biased surface flux estimates. These studies performed pseudo-data experiments or used a small number of observations, and focused mostly on the impact of different PBL physics schemes. There is agreement among the studies that

misrepresentation of vertical mixing causes biases in ML $CO_2$ mole fractions, and that these biases directly affect inverse flux estimates. Vertical mixing, however, is not solely affected by the PBL parameterization. Therefore, investigations of vertical mixing of $CO_2$ remain incomplete. Additional parameterizations that impact the transport of air masses both horizontally and vertically should be evaluated.

In this work, we study uncertainty in an atmospheric transport model using a multi-physics approach not limited to the

evaluation of the PBL schemes and LSMs. This evaluation will include different LSMs, cumulus parameterizations (CP), microphysics parameterizations (MP), and initial and boundary conditions used by the WRF model. We will evaluate model performance using observations of atmospheric transport variables, PBL depth, wind speed and wind direction, expected to be most important to ML $CO_2$ mole fractions. We aim to quantify the uncertainty of the atmospheric transport model and propagate these errors into the $CO_2$ mole fractions. We will focus on the following questions: How do different physical



parameterization schemes affect ML $CO_2$ mole fractions? Are some physics parameterizations more effective/accurate than others at simulating atmospheric conditions important to interpreting $CO_2$ mole fraction observations in the PBL? What are the nature and magnitude of random and systematic errors in the WRF model, and how does this depend on model configuration? We will address these questions by exploring atmospheric transport model performance over a large, densely

instrumented region, the US Midwest, site of the Mid-Continental Intensive (MCI) study (Ogle et al., 2006). Evaluating the atmospheric transport during summer, the most biologically active time of the year, is a first step toward a more rigorous and complete atmospheric inversion that quantifies random transport errors more accurately, and minimizes transport biases. This work will expand our ability to assess, understand, and reduce transport errors in future atmospheric inversions.

## 2 Methods

### 2.1 Region

The region selected for our study is the Midwest region of the United States (Figure 1). The Midwest of U.S. was chosen because the first multi-year (2007-2009) campaign with a high-density $CO_2$ measurement network was deployed in this region (Ogle et al., 2006, Miles et al., 2012). This field campaign, part of the North American Carbon Program (NACP), was called the Mid-Continental Intensive (MCI) and encompassed the agricultural belt in the north-central U.S. The MCI

campaign is unique for its density of well-calibrated (Richardson et al., 2012) atmospheric $CO_2$ mole fraction measurements intended to constrain the region's carbon budget. We describe the operational rawinsonde and GHG tower networks over the region in Section 2.2.4. These networks provided significant observational constraint on both transport and GHG mole fractions, which allow us to evaluate and quantify the atmospheric transport errors in this study.

### 2.2 Atmospheric model setup

The atmospheric transport model used in this study to generate our 45-member physics ensemble is the Weather Research and Forecasting (WRF) model version 3.5.1 (Skamarock et al., 2005) and a modified chemistry module for $CO_2$ (called WRF-ChemCO2, Lauvaux et al., 2012). The atmospheric column in each simulation is described with 59 vertical levels, with 40 of them within the first 2-km of the atmosphere. Two nested domains were used. The coarse domain (d01) uses a horizontal grid spacing of 30-km and the nested or inner domain (d02) uses 10-km grid spacing (Figure 1). The coarse

domain covers most of the United States and parts of Canada and the nested domain is centered over Iowa and covers the Midwest region of United States. The nesting method employed is the "one-way" nesting in which the outer domain constrains the inner domain through nudging of the boundary conditions that drive the meteorology once the outer domain simulation has finished (Soriano et. al., 2002). No feedback from the inner domain to the coarse domain was allowed. For our sensitivity study, only the inner domain (d02) has been analyzed as it covers the area of interest.



### 2.3 Ensemble configuration

Similar to any domain-limited atmospheric model, transport errors arise from initial and boundary conditions and the different physics parameterizations. Therefore, we have built an ensemble of 45-members using different physical parameterization schemes and large-scale initial and boundary conditions from reanalysis products (see Table 1). WRF

offers multiple options for the LSM, PBL, cumulus, and microphysics schemes. The members in our multi-physics ensemble all use the same radiation schemes (both long wave and shortwave) but the land surface, surface layer, boundary layer, cumulus, and microphysics schemes are varied for both the inner and the outer domain. In addition, we have initialized the meteorological boundary and initial conditions with different datasets. Table 2 shows the different options used in this study.

### 2.4 Physics parameterization schemes

*a. Land surface models (LSMs)*

The land surface models (LSMs), which ingest land-surface properties, soil, and surface conditions from driver data, simulate the conditions at the land surface, including surface energy fluxes. The partitioning of these fluxes affects the structure and depth of the PBL through the turbulence parameterization, hence modifying the near-surface in situ $CO_2$ mole

fractions. To evaluate the sensitivity of modelled mole fractions to the surface conditions, three LSM schemes are chosen for this study: the 5-layer soil thermal diffusion model (Dudhia, 1996), the Noah land surface model (Chen and Dudhia, 2001), and the Rapid Update Cycle (RUC) (Smirnova, 2000). These LSMs differ in several aspects, from the description of soil properties to the physical processes driving the land-surface interactions. The thermal diffusion model uses a simple thermal diffusion equation to transfer thermal energy from the ground to the atmosphere, describing the belowground profile with 5

soil layers (Dudhia, 1996). This LSM also includes snow-covered land and constant soil moisture values for a given land use type and season. The Noah LSM scheme uses time-dependent soil temperature and moisture for four soil layers, canopy conductance and moisture, and snow cover prediction (Chen and Dudhia, 2001). The RUC LSM scheme includes six soil layers and includes the effects of vegetation, canopy water, and snow (Smirnova, 2000). This scheme also includes parameterizations for snow and frozen soil (Smirnova, 2000).

*b. Planetary boundary layer (PBL) schemes*

The planetary boundary layer (PBL) is directly influenced by frictional drag, sensible heat flux, and evapotranspiration, all of which are responsible for generating turbulent eddies. The PBL schemes parameterize turbulent vertical fluxes of heat, momentum, and moisture within the PBL and throughout the atmosphere. The three PBL schemes used in this study are the

Yonsei University (YSU) (Hong et al., 2006) PBL scheme, the Mellor-Yamada-Janjic (MYJ) (Janjic, 2002) PBL scheme, and the Mellor-Yamada-Nakanishi-Niino (MYNN) PBL scheme (Nakanishi & Niino, 2004). These three PBL schemes differ in the treatment of turbulent diffusion. The YSU scheme is a first order scheme that includes non-local eddy diffusivity


coefficients to compute turbulent fluxes. The YSU scheme explicitly calculates entrainment at the top of the PBL as a function of the surface buoyancy flux. The MYJ and MYNN 2.5 PBL schemes are local closure schemes that include a prognostic equation for turbulent kinetic energy (TKE) and a level 2.5 turbulence closure approximation to determine eddy transfer coefficients. The MYJ scheme implicitly calculates the entrainment layer while the MYNN uses a more explicit

representation of entrainment at the top of the PBL (Román-Cascón et al., 2012). The MYNN 2.5 is a variation of the MYJ PBL scheme that includes a nonlocal component of the turbulent mixing that reduces potential cold biases and increases PBL depths. The MYJ PBL scheme used in this study has been slightly modified to allow for very low turbulence regimes (e.g. nocturnal stable conditions) with a decreased minimum value for TKE.

*c. Cumulus parameterizations*

The cumulus parameterization (CP) schemes are used with the aim of representing the vertical fluxes due to unresolved updraft and downdrafts and compensating motion outside the clouds. In this study we use two different cumulus parameterization schemes, Kain-Fritsch (KF) (Kain, 2004), Grell-3D (G3D) (Grell and Devenyi, 2002). The KF scheme is deep and shallow convection sub-grid scheme, which uses a simple cloud model that simulates moist updrafts and

downdrafts along with detrainment and entrainment effects. The G3D cumulus scheme is based on the Grell (1993) scheme and G3D is a scheme for higher resolution domains allowing for subsidence and neighboring columns. The G3D uses a large ensemble of closure assumptions and parameters that are used in numerical models and implements statistical techniques to determine the optimal value for feedback to the entire model (Pei et al., 2014). The cumulus parameterization is theoretically only valid for coarse grid resolutions (e.g., greater than 10 km) and should not be used when the model has a higher

resolution (e.g., less than 5km) and will resolve cumulus convection (Skamarock et al., 2005). Therefore, we are in a 'grey-zone' (5-10km), where it is unclear if cumulus parameterization should be used or not. For that reason, we also ran simulations that do not use a cumulus parameterization scheme in the nested domain.

*d. Microphysics parameterizations*

Microphysics parameterizations (MP) describe cloud and precipitation processes. In this study we use two MP schemes: the WRF Single-Moment 5-class (WSM5) scheme (Hong et al., 2004) and the Thompson scheme (Thompson et al., 2004). The WSM5 scheme is a single moment parameterization that includes five species: water vapor, cloud water, cloud ice, rain, and snow, which are all treated independently. The Thompson scheme is a double moment scheme, which predicts the mole fraction of five hydrometeors species, the number concentration of ice phase hydrometeors, and rain.

**2.5 Meteorological initial and boundary conditions**

Two meteorological datasets provide the initial and lateral boundary conditions for our regional model. For initialization, WRF interpolates the coarse-resolution analysis products onto the model grid and calculates the values of the parent domain





lateral boundaries. The inner grid uses the boundary conditions of the parent domain. In this study, we compare two different meteorological datasets: the North America Regional Reanalysis (NARR) (Mesinger et al., 2006), and the Final Operational Global Analysis (FNL). The NARR dataset was developed at the Environmental Modeling Center (EMS) of the National Centers for Environmental Prediction (NCEP). NARR uses a high resolution NCEP Eta Model with a horizontal grid spacing

of 32 km and includes 45 vertical levels. NARR provides both initial and boundary conditions at 3-hourly intervals. The NCEP FNL analysis data has a horizontal grid spacing of 1°×1° and is prepared operationally every six hours. The FNL is prepared with the same model that NCEP uses in the Global Forecast System (GFS). The initial conditions in the WRF simulations are reset every 5 days to avoid the growth of model errors in the absence of data assimilation. The WRF model spin-up takes about 18 hours, so we use model results after 18 hours of the first day of each 5-day simulation segment. We

compared model-model differences over the 5 days and found no significant trend over the 5-day periods once removing the first 18 hours of spin-up.

**2.6 CO$_2$ surface fluxes**

For this study, we used the summer 2008 posterior surface fluxes from the data assimilation system CarbonTracker[1] version 2009 (CT2009) (Peters et al., 2007). This system produces CO$_2$ flux estimates by integrating daily daytime averaged CO$_2$

mole fractions from continuous hourly observations and then minimizing the differences between the observed and modeled atmospheric CO$_2$ mole fractions. The Transport Model 5 (TM5) offline atmospheric tracer transport model (Krol et al., 2005) driven by the European Centre for Medium-Range Weather Forecasts (ECMWF) operational forecast model, propagates the surface fluxes to generate 3D mole fractions of CO$_2$ across the globe.

The CO$_2$ surface fluxes are represented by different sub-components, which include: fossil fuel emissions, biomass burning,

terrestrial biosphere exchange, and ocean-atmosphere exchanges. The annual fossil fuel emissions used in CT2009 are from the Carbon Dioxide Information and Analysis Center (CDIAC) (Boden et al., 2009). These fossil fuel fluxes are mapped onto a 1°×1° grid and are then distributed into country totals according to the spatial patterns from the EDGAR-4 inventories (Olivier and Berdowski, 2001). Biomass burning is based on the Global Fire Emission Database version 2 (GFEDv2). The dataset consists of 1°×1° gridded monthly burned areas, fuel loads, combustion completeness, and fire emissions. Prior

terrestrial biosphere flux estimates come from the Carnegie-Ames Stanford Approach (CASA) global biogeochemical model (van der Werf et al., 2006; Giglio et al., 2006) with three-hour variability imposed by temperature and incoming radiation (Olsen and Randerson, 2004). The CASA biosphere model produces net primary production (NPP) and heterotrophic respiration fluxes with a monthly time resolution at 0.5°×0.5° spatial resolution. The long-term ocean fluxes and uncertainties are derived from inversions reported in Jacobson et al., (2007). Ocean inverse flux estimates are composed of

preindustrial (natural), anthropogenic flux inversions, and an additional level of biogeochemical interpretations (Gloor et al.,

---

[1] http://carbontracker.noaa.gov





2003; Gruber, Sarmiento and Stocker, 1996). Similar to most $CO_2$ inverse systems, the fossil fuel and fire emissions are specified (i.e. remain constant) and only the oceanic and terrestrial biosphere fluxes are optimized.

**2.7 Dataset**

Our interest is to explore and quantify atmospheric transport errors over the Midwest U.S. using observations that we have
over this region. Therefore, we will evaluate the errors over the inner domain (D2) of our models. Figure 1 shows the location of all the stations that provide atmospheric $CO_2$ mole fractions and the meteorological observation sites that will be used. Meteorological data were obtained from the University of Wyoming's online data archive (http://weather.uwyo.edu/upperair/sounding.html) for the 14 rawinsonde stations shown in Figure 1. In-situ atmospheric $CO_2$ mole fraction data are provided by gas analysers operating continuously on seven communication towers (Figure 1) (Miles et
al., 2012). Five of these towers were part of an experimental network, deployed from 2007 to 2009 (Richardson et al., 2012; Miles et al., 2012, 2013; http://dx.doi.org/10.3334/ORNLDAAC/1202). The other two towers (Park Falls-WLEF and West Branch-WBI) are part of the Earth System Research Laboratory/Global Monitoring Division (ESRL/GMD) tall tower network (Andrews et al., 2014; https://www.esrl.noaa.gov/gmd/ccgg/insitu/). Each of these towers sampled air at multiple heights, ranging from 11 m AGL to 396 m AGL.

**2.8 Data selection**

Most atmospheric inversions that use continental PBL observations only use daytime $CO_2$ mole fractions from continuous observations (Law et al., 2003), with the exception of mountain sites whose nighttime data is thought to sample free tropospheric conditions (Brooks et al., 2012). Only daytime measurements are assimilated due to the difficulty in simulating strong vertical gradients in the nocturnal boundary layer.  Vertical gradients are minimized during daytime under well-mixed
boundary layer conditions (Bakwin et al., 1998). Therefore, both models and observations will be evaluated during daytime. We analyzed $CO_2$ mole fractions collected from sampling levels at or above 100m AGL, which is the highest observation level available across the entire MCI network (Miles et al., 2012). This ensures that the observed mole fractions reflect the influence of regional $CO_2$ fluxes and are minimally influenced by near-surface gradients of $CO_2$ in the atmospheric surface layer (ASL) due to local $CO_2$ fluxes (Wang et al., 2007). Both observed and simulated $CO_2$ mole fractions are averaged from
1800 to 2200 UTC (12:00-16:00 LST), the daytime period when the boundary layer should be convective and the $CO_2$ profile well mixed (e.g., Davis et al., 2003; Stull, 1988). This averaged mole fraction will be referred to hereafter as the daily daytime average (DDA).

In this study, we will also evaluate the PBL wind speed (hereafter wind speed), PBL wind direction (hereafter wind direction), and PBL height (PBLH) from the different rawinsonde stations. Similar to the $CO_2$ mole fractions, we want our
meteorological observations to be within the well-mixed layer. Therefore, we use the wind speed and wind direction observed approximately 300 m above ground level (AGL). $CO_2$ mole fraction observations were sampled at about 100m, however, the availability of meteorological observations at this height is too low to collect a sufficient number of data for our



statistical evaluation. The observed PBLH was estimated using the virtual potential temperature gradient with a threshold of 0.2 K/m. We want our simulated meteorological variables to be close to the observational level, therefore we use wind speed and wind direction from level 11 (~350 m) of the model. The WRF model provides an estimate of the PBLH, but the methodology used to diagnose these values varied with the PBL scheme used in the simulation. To remain consistent, we

decided to calculate the PBLH in WRF with the same potential temperature gradient method that is used for the rawinsonde data. Rawinsonde stations across this region collect data at 1200 UTC and 0000 UTC, but our model-data evaluation will be done for daytime conditions only. Therefore, both the modeled results and data will be evaluated in the late afternoon (i.e., 0000 UTC) corresponding to well mixed conditions.

### 2.9 Evaluation methodology or analyses of the models

Comparison to measurements of wind speed, wind direction, PBLH, and DDA $CO_2$ mole fractions are used to inform the performance of each model configuration. Modeled data are extracted from the simulations using the nearest grid points to the locations of our observations. Each model configuration is evaluated from June 18 to July 21, 2008 for the meteorological variables and from June 26 to July 22, 2008 for the $CO_2$ mole fractions. Summer in the U.S. Midwest

corresponds to the peak of the growing season for both crops and most non-agricultural ecosystems (except grasslands). We focus here on the growing season because the large biogenic fluxes make this period the most important time of year for understanding the relationship between fluxes and $CO_2$ mole fractions. We first explore meteorological variables, and the sensitivity of $CO_2$ to atmospheric transport but without comparison to observations, to avoid confounding the impact of transport with errors from $CO_2$ surface fluxes and CarbonTracker global $CO_2$ mole fractions. Finally, we compare to $CO_2$

observations with the knowledge that the results include both transport and $CO_2$ flux errors.

#### a. Analyses of physics parameterization and reanalyses impact

The daily mean of root mean square difference (RMSD) among ensemble members is used to isolate the atmospheric transport variability and evaluate the impact of the physics parameterizations on both $CO_2$ mole fractions and PBL dynamics. The RMSD does not consider the observations as we take the square root of the average difference between model

configuration and the ensemble mean,

$$\overline{RMSD} = \sum_{i=1}^{N} \frac{\sqrt{\frac{1}{n}\sum_{j=1}^{n}(p_{ji}-\mu_i)^2}}{N} \qquad (1)$$

where $p_{ji}$ is the predicted variable for ensemble member $j$ and day $i$, $\mu_i$ is the mean of the ensemble for day $i$, $N$ is the total number of days, and $n$ is the number of members. The RMSD was estimated for the different physics parameterization used

(i.e., LSM, PBL schemes, CP, MP) and reanalysis. The RMSD of the simulated $CO_2$ mole fractions was used to explore if other physics parameterizations have a significant impact on $CO_2$ mole fractions compared to the PBL parameterizations. To





explore which parameterizations impact the PBL dynamics we applied the RMSD to the three selected meteorological variables (i.e., PBLH, wind speed, and wind direction), assuming these variables contribute the most to the representations of the $CO_2$ mole fraction distributions in the PBL. The RMSD for the meteorological variables were then averaged across all of the rawinsonde sites. The RMSD of the $CO_2$ mole fractions was estimated using the simulated $CO_2$ mole fraction at each

communication tower and then averaged across the tower sites to match the model-data residual.

#### b. Analyses of model-data residuals

A series of statistical analyses are used to assess the performance of the different model configurations for the three meteorological variables wind speed, wind direction and PBLH. The different metrics used include the root mean square

error (RMSE) and mean bias errors (MBE),

$$RMSE = \sqrt{\frac{1}{N}\sum_{i=1}^{N}(p_i - o_i)^2},$$    (2)

$$MBE = \frac{1}{N}\sum_{i=1}^{N}(p_i - o_i),$$    (3)

where $o_i$ is the observed variable for day $i$, $p_i$ is the predicted variable for day $i$, and $N$ is the total number of days. The RMSE

represents the magnitude of the model error without regard to the long-term mean (Wilks, 2011). The MBE describes the model-observations difference averaged errors over the entire period (Wilks, 2011), and identifies model bias. These two metrics are critical to inverse flux estimates as biases can arise from day-to-day (which we will refer as random) or longer-term (systematic) errors in the transport model. We acknowledge that the propagation of meteorological errors to mole fractions, and mole fractions errors to surface fluxes is complex, but these metrics provide valuable insight into model

performance. Each of these statistics (i.e., RMSE and MBE) was estimated for each model and each rawinsonde site using the late afternoon (0000 UTC) soundings.

Finally, we compare modelled and simulated PBL $CO_2$. We use our different model configurations, which all share the exact same surface fluxes and identical boundary conditions to explore the impact of the transport errors on $CO_2$ mole fractions.

We present the impact of model configurations on the DDA $CO_2$ mole fraction model-data mismatches (or residuals) with Taylor Diagrams and correlation between model-data residuals in meteorological variables and DDA $CO_2$ mole fractions. The Taylor diagram relies on three nondimensional statistics: the variance ratio (model variance normalized by the observed variance), the correlation coefficient, and the normalized centered root-mean-square (CRMS) difference (Taylor, 2001). The variance ratio or normalized standard deviation (NSD) indicates the difference in amplitude between the model and the

observation. If this ratio is less than 1.0, then the model tends to underestimate the amplitude compared to the observation. The correlation coefficient measures the similarity in the temporal variations between the model and the observation,




regardless of the amplitude. This correlation coefficient has a range of $-1.0 \leq R \leq 1.0$ and is insensitive to systematic errors. As R approaches 1.0, the model approaches agreement with the observation. The CRMS is normalized by the observed standard deviation and quantifies the ratio of the amplitude of the variations between the model and the observation. The CRMS is also insensitive to systematic errors. Temporal correlations between the modeled-observed residual in

meteorological variables and $CO_2$ mole fractions are used to determine the impact that meteorological errors have on the PBL $CO_2$ mole fractions. This model-data correlation will be done between each $CO_2$ observing site and rawinsonde site, therefore we will be able to observe if any correlation is dependent on the distance between sites. The model-data residual includes both flux and transport errors, therefore, these errors will not show the accuracy of the transport model. Nevertheless, each simulation uses the same $CO_2$ flux and boundary conditions that allows us to use the model-data residuals

as an indicator of the differences between model configurations.

### 3. Results

### 3.1 Impact of physics parameterizations on atmospheric $CO_2$ mole fractions

The daily mean of root mean square difference (RMSD) of the simulated $CO_2$ mole fraction was used to explore the sensitivity of $CO_2$ mole fractions to model physics parameterization and meteorological reanalysis. The RMSD was

computed for different parameterizations schemes (i.e., LSM, PBL, CP and MP) and for two reanalysis products (i.e., NARR and FNL).   For each group of parameterizations, the model configuration remained identical except for the tested parameterization scheme. For example, to evaluate the impact of LSM schemes on $CO_2$ mole fractions, three LSM schemes were used while preserving the exact same physical schemes for the PBL, CP, MP, and the re-analysis data. Figure 2 shows the results of these experiments. $CO_2$ mole fraction RMSD is greatest for the LSM, followed by the PBL scheme and CP.

The microphysics parameterization has the least impact on $CO_2$ mole fractions. Only two microphysics parameterizations are tested in this ensemble but additional tests using only two options for all the different physic parameterizations produced similar results.

We also explore how much the variability in PBL winds and depth are influenced by physics parameterizations. Figure 3 shows the RMSD of PBL wind speed and direction, and PBLH over the entire simulation period. The results for all three

meteorological variables are similar to those for $CO_2$ mole fractions. Reanalysis has a greater impact on wind speed (Figure 3b) and wind direction (Figure 3c) than it does on PBLH (Figure 3a). It is worth noting that the PBLH RMSD (Figure 3a) shows the same RMSD ranking (i.e. relative importance of the physics) as for $CO_2$ mole fraction RMSD (Figure 2).

Based on the evaluation of the $CO_2$ mole fraction, wind speed, wind direction, and PBLH RMSD, the LSM has the greatest impact on PBL $CO_2$ transport, followed closely by the PBL scheme, CP and reanalysis. All the parameterization schemes,

including the reanalysis data source, have a significant impact on each of these variables.   The RMSDs were significant values compared to typical spatial and temporal differences (for PBL $CO_2$, see Miles et al., (2012)) and for mean PBL properties (PBLH, winds), confirming the importance of model parameterization on these variables.





### 3.2 Meteorological day-to-day variability

Figure 4 shows a time series of the 0000 UTC observed and simulated wind speed (Figure 4a), wind direction (Figure 4b) and PBLH (Figure 4c) from June 18 to July 21, 2008 at the Chanhassen, MN (MPX) rawinsonde site. Across the study region, we found maximum monthly average model-data differences across sites and configurations of 9 m/s for wind speed,

153 degrees in wind direction and 2000 m for PBLH. These values confirm the large spread among model results and sites over the simulation time period. Other sites have similar characteristics to Figure 4. The ensemble shows less variability (i.e., relative spread of the ensemble compared to the observed variability) for the wind speed and wind direction compared to the PBLH. The time series at each rawinsonde site shows that for certain days, all ensemble are biased (i.e., all the members either overestimate or underestimate) as compared to observed wind speed and wind direction (e.g. DOY 181 and 201,

respectively). The time series of the PBLH, however, shows that simulated PBLH can vary significantly across the different physics configurations and that the ensemble encompasses the observed PBLH over the time period.

### 3.3 Characterization of transport errors

### 3.3.1 Root mean square error (RMSE)

Figure 5 shows the regionally and monthly averaged RMSE of wind speed (Figure 5a), wind direction (Figure 5b), and

PBLH (Figure 5c) for the different model configurations. For both wind speed and wind direction, we found small to no differences in the regional RMSE as a function of model configuration. Although the regional RMSE for both wind speed and wind direction are fairly constant, the two variables have the same two model configurations with the highest RMSE. These two configurations share the same LSM scheme (RUC) and the same PBL scheme (MYJ) (models 14 and 23 see Figure 5a,b and Table 1). Differences among configurations are larger in the regional RMSE of the PBLH (Figure 5c), with

configuration RMSEs ranging from 680m to 1149m. The model configurations that show the highest PBLH RMSE include the same LSM (RUC) and PBL parameterization scheme (YSU) (models 4, 13, 22 and 34 see Figure 5c and Table 1). Although the configurations that shows the highest RMSE are not always the same across the different variables, these configurations share the same LSM (RUC). The two model configurations that showed the lowest RMSE for both wind speed and wind direction both used MYNN 2.5 as their PBL parameterization. Many configurations show low RMSE for the

PBLH and all the configurations with low RMSE use either the MYJ scheme or the MYNN 2.5 scheme. However, no single configuration performs best at the regional scale for all of the meteorological variables.

We computed the ensemble mean of the monthly averaged RMSE at each of the rawinsonde sites for wind speed (Figure 6a), wind direction (Figure 6b) and PBLH (Figure 6c). We did not find any regional patterns in wind speed (Figure 6a) and wind

direction (Figure 6b). However, PBLH shows that the highest RMSEs are located in the west of the domain, with an RMSE 400 m or higher than the sites in the East.





Figure 7 shows the monthly average RMSE of wind speed (Figure 7a-c), wind direction (Figure 7d-f), and PBLH (Figure 7g-i) for each model configuration at specific rawinsonde sites. Although the RMSE was computed at each of the rawinsonde sites, we show only three sites located in three different regions of the domain: LBF in the west (Figure 7a, d, g), MPX which is close to the center of the domain (Figure 7b, e, h), and APX in the eastern part of the domain (Figure 7 c, f, i).

Similar to the regional RMSE (Figure 5), both LBF and MPX shows that the LSM RUC leads to the highest RMSE for the three meteorological variables. However, this pattern is not found at APX, where other configurations show the highest RMSE for wind speed, wind direction and PBL height. Across simulations and meteorological variables, RMSEs vary but no configuration shows a lower value across all sites.

### 3.3.3 Mean bias Error (MBE)

The average over-or-underestimation of the model configurations is assessed by computing the regional monthly average MBE for wind speed (Figure 8a), wind direction (Figure 8b) and PBLH (Figure 8c). In this study, a positive MBE means the model configuration is systematically higher than the observation. We found remarkable variations in the regional MBE both as a functions of different model configurations and across the meteorological variables. The regionally averaged PBL wind speed bias for any single ensemble member ranges from -0.2 to 1.2 m/s, relative to the mean regional midday wind speed of

6.2 m/s, showing that the bias of any single ensemble member ranges from less than 5% to nearly 20% of the regional mean PBL wind speed (Figure 8a). All configurations that use YSU (e.g., models 1, 4, 7, 10 see Figure 8a and Table 1) have greater regional wind speed biases than the rest of the PBL schemes. The regional MBE for wind direction varies according to model configuration. Models using YSU as PBL schemes tent to show a systematic positive bias in the wind direction (e.g., models 1, 4, 7, 10 see Figure 8b and Table 1), whereas models that use MYJ as PBL scheme show a negative bias (e.g.,

models 2, 5, 8, 11 Figure 8b and Table 1). Similar to the wind direction, the regional PBLH bias is correlated with model configuration. Any model configuration that uses YSU shows a positive bias, larger than the rest of the PBL schemes (e.g., models 1, 4, 7, 11 see Figure 8c and Table 1). The model configurations that do not include cumulus parameterizations (white filled bars; Figure 8c) also show positive biases, with one exception, regardless of the choice of LSM or PBL scheme used. The wind speed analysis shows that the two model configurations with the smallest regional MBE (± 0.1 m/s) share the

same LSM (Thermal Diffusion) and PBL (MYNN 2.5) parameterization. For wind direction, two of the three model configurations with the lowest MBE (± 0.1 degrees) use the same LSM (Noah) and PBL (YSU) parameterization. All 15 model configurations with the lowest MBE for PBLH (± 100 m or less) share the same PBL parameterizations (MYJ and MYNN 2.5). Although the configurations that provide the lowest regional MBE is not the same across all variables, we found that the lowest biases for the three variables were produced by model 18 (see Table1). This model configuration is

driven by the NARR reanalysis product, and used Thermal Diffusion as LSM, MYNN as PBL scheme, Grell-3D as CP and WSM 5-class as MP.



The spatial structures of the MBE over a month are evaluated by estimating the ensemble mean of the MBE at each rawinsonde site (Figure 9). The ensemble mean of the MBE reveals a spatial pattern in the wind speed (Figure 9a) and PBLH (Figure 9c). The map of wind speed MBE (Figure 9a) shows that the ensemble is positively biased in the eastern region of the domain. However, sites in the western region of the domain show that the ensemble average has either negative

or near-zero wind speed MBEs. The PBLH MBE map (Figure 9c) also shows a clear spatial pattern, with the highest values, nearly all positive, at sites located in the western part of the domain, whereas the sites in the eastern part of our domain show a smaller MBE and no distinct regional sign. PBL wind direction does not show any spatial pattern in the ensemble mean of the MBE (Figure 9b). We found that our ensemble of simulations can produce an MBE range from rawinsonde site to rawinsonde site of ±1.5 m/s in wind speed, ±20 degrees for wind direction, and ± 400 m for PBLH.

Figure 10 shows the MBE of three sites. The analysis was performed for all the sites (not shown) and site representative of regional patterns were chosen. The three sites shown are located in three different regions of the domain: ABR in the west (Figure 10a, d, g), DVN which is close to the center of the domain (Figure 10b, e, h), and BNA in the eastern part of the domain (Figure 10c, f, i). Most of the model configurations shows positive wind speed MBE (overestimation) for the

majority of the rawinsonde sites (e.g., Figure 10b-c), however, one site shows both positive and negative MBE for the different model configurations (e.g., Figure 10a). Overall we found that 10 out of the 14 rawinsonde sites show all the model configurations with a positive wind speed bias; these sites were located in the eastern and center areas of the domain. However, the MBEs for wind direction (e.g., Figure 10d-f) and PBLH (e.g., Figure 10g-i) are highly variable across the rawinsonde sites. At the majority of the sites, the simulations had both positive and negative biases. Although wind speed

and wind direction do not show any of the simulations with a systematic behaviour across the sites, PBLH MBE showed some simulations with systematic bias across the different sites. The highest positive biases were found in configurations that use RUC as the LSM and YSU as the PBL scheme in the western region of the domain (e.g., Figure 10g, red bars). This is unlike the eastern region of the domain, where the highest biases were dominated by configurations that use Thermal Diffusion as the LSM and YSU as the PBL scheme (e.g., Figure 10i, white bar with green border). These results indicate that

wind speed MBEs are strongly impact by other components of the model (e.g., reanalysis data set) or that the WRF transport model carries a systematic bias that will show up regardless of the configuration used. However, PBLH bias is highly controlled by two components of the model, the LSM and the PBL parameterization scheme. Overall, the spatial patterns show that no configurations can avoid spatial biases across the region.

### 3.4 Sensitivity of $CO_2$ mole fractions to model configuration

Figure 11 shows simulated and observed atmospheric DDA $CO_2$ mole fraction for Centerville (Figure 11a) and Kewanee (Figure 11b) from June 26 to July 21, 2008. For this period, both sites show large residuals that are not encompassed by the ensemble spread (RMSD) for several periods (e.g. DOY 182-183 at Centerville or DOY 185-186 at WBI). This result



suggests that transport model errors from our ensemble only represent a fraction of the total uncertainty in our modelling system. Additional errors can be due to incorrect $CO_2$ surface fluxes and boundary conditions. Over the region, most of the sites show that the ensemble generally underestimates the atmospheric $CO_2$ mole. We note here that this ensemble has not been calibrated, therefore the ensemble spread is unlikely to serve as quantification of WRF transport errors or total error in

simulated PBL $CO_2$, but this sensitivity test could have resulted in an ensemble spread that is much larger than the model-data differences. Our results suggest that the spread of this physics ensemble underestimates total model-data error in PBL $CO_2$.

To evaluate the performance of the different models over the month, we computed the correlation coefficient, the NSD and

the CRMS difference (Taylor, 2001) for each of the in-situ sites. These results are presented as Taylor Diagrams (Figure 12) using the DDA observed and simulated $CO_2$ mole fractions. Nearly all ensemble members overestimate the temporal variability at in PBL $CO_2$ (e.g., Figure 12a) and at some sites all members overestimate the temporal variability (e.g., Figure 12b). The correlation between simulated and observed $CO_2$ mole fraction can vary from 0.8 to 0.1, indicating a wide range of model performances at site-level. Interestingly, some of the models that show a high correlation between the modeled and

observed DDA $CO_2$ mole fractions are the model configurations with the highest PBLH bias (see Figure 8c, model 4 and 22).

The correlation between meteorological and $CO_2$ mole fraction model-data differences is evaluated using the MBE for each model at the different rawinsonde and $CO_2$ tower sites. These correlations (Figure 13) reveal that to first order, errors in

simulated PBLH govern model-data differences in PBL $CO_2$ mole fraction. Both wind speed (Figure 13a) and wind direction (Figure 13b) shows low correlations, whereas PBLH (Figure 13c) shows consistently positive correlation with the $CO_2$ mole fraction errors across all sites. We did not find any relationship between error correlation and distance. These results suggest that the bias errors in the in situ $CO_2$ mole fractions are directly related to the MBE in PBLH. The sign of the correlation (overpredicted PBLH correlated with overpredicted PBL $CO_2$) is expected given net uptake of $CO_2$ by the regional

biosphere.

**4. Discussion**

The evaluation of the RMSD of daytime PBL $CO_2$ mole fractions shows that all the physics parameterizations have a significant impact on the simulated values, with only the microphysics parameterization showing a lesser impact (Figure 2). Previous research has focused on the potential impact of PBL schemes on $CO_2$ mole fractions (e.g. Kretschmer et al., 2012,

2014; Lauvaux and Davis 2014). Results from our study indicate that other physics parameterizations including the LSM and CP generate errors of similar magnitude in simulated daytime PBL $CO_2$ mole fractions (Figure 2). The PBLH is also sensitive to all of these physical parameterizations (Figure 3a), and there is a high correlation between PBLH errors and $CO_2$ mole fraction errors (see Figure 13c). In this sense, our results agree with previous research that assumes that the



misrepresentation of the PBLH plays an important role in PBL $CO_2$ errors (Stephens et al., 2007; Gerbig et al., 2008; Kretschmer et al., 2012). We show, however, that multiple elements of the modeling system, not just the PBL parameterization, influence PBLH. Further, although PBL wind speed (Figure 13a) and wind direction (Figure 13b) errors are not clearly correlated with PBL $CO_2$ errors, this does not imply that these errors are unimportant. Figure 2 also shows

that the reanalysis has an impact on atmospheric $CO_2$ mole fractions, which indicates that even if the wind speed and wind direction errors do not show a high correlation with atmospheric $CO_2$ errors (Figure 13a-b), these two variables can contribute to the errors in $CO_2$ mole fractions. Indirectly, we demonstrated that the reanalysis directly impacts $CO_2$ mole fractions by changing wind speed and direction in WRF (Figure 3b-c), whereas PBLH errors are primarily driven by physical schemes (surface and PBL schemes). The relationship between PBL winds and $CO_2$ mole fraction is dependent on

the local spatial distribution of $CO_2$ surface fluxes and could easily show no clear correlation when averaged over time and space. However, we know errors in these two variables can impact the distribution and magnitude of the inverse $CO_2$ fluxes over the region.

The square root of the model errors (RMSE) of wind speed and wind direction show similar magnitudes in the errors

regardless of the model configuration that we use. Additionally, we found over the region a systematic positive bias (MBE; overestimation) of wind speed for the majority of the model and sites. These results specially the ones from wind speed lead us to consider other elements of the models as a contributor of the errors. Past literature has stated and shown that the WRF mesoscale model has a tendency to produce high wind speed over land (Cheng and Steenburgh, 2005; Roux et al., 2009; Zhang et al., 2009; Yerramilli et al., 2010; Jimenez and Dudhia, 2012). Some studies have attributed this high wind speed

bias to the smoothed representation of the topography (Jimenez and Dudhia, 2012; Santos-Alamillos et al., 2013). Biases of approximately 3 m/s have been attributed to this misrepresentation of topography. Fovell and Cao (2014) argue that the misrepresentation of the terrain, or possibly the vegetation, can produce a biased roughness length that can lead to wind speed biases of about 2 m/s. There are other factors that contribute to wind speed errors such as the reanalysis product (Figure 3.b). We recommend further analysis of the WRF model and its driver and input data to better understand PBL wind

speed random and systematic errors.

The PBLH biases across the region shows that YSU PBL schemes tends to produce higher PBLH than MYJ PBL schemes. This is consistent with previous studies (Hu et al., 2010, Coniglio et al., 2013; Milovac et al., 2016) that have found local PBL schemes (MYJ and MYNN) producing shallower PBLHs compared to nonlocal PBL schemes (YSU). As explained in

Coniglio et al., (2013), the MYJ scheme produces cool and moist conditions near the ground and hence low vertical mixing, whereas the YSU scheme produces warm and dry conditions in the PBL resulting in deep mixing. Since daytime PBLH is closely linked to the surface energy balance, an additional analysis was performed using the sensible heat fluxes observed at eddy covariance stations from the AmeriFlux network (Boden et al. 2013; http://ameriflux.lbl.gov). The sensible heat flux was averaged from 1200 to 2300 UTC and we computed the MBE of the sensible heat flux for the eddy covariance stations



close to the rawinsonde sites. We found that model configurations that use YSU as the PBL scheme in combination with RUC (Figure 14c,d, red bars) or Thermal Diffusion (Figure 14a,b, green bars) as the LSM have the highest positive bias for sensible heat flux (Figure 14c,d), consistent with the positive biases in PBLH associated with these configurations.

We also found a spatial pattern in PBLH bias averaged over all model configurations (Figure 9c), where the West region of the domain shows a large positive bias and with no persistent bias in the East part of the domain. This spatial gradient may be associated with the representation of warmer and drier areas in the west and cooler and moister areas in the eastern portion of the domain (Molod et al., 2015). This spatially-structured PBLH bias could be associated with the choice of LSM since high biases are dominated by members using Thermal Diffusion scheme (e.g., Figure 10i, white bar with green border)

in the East region of the domain and by members using RUC (e.g., Figure 10g, red bars) in the West. Although, both RUC and Thermal Diffusion LSM tend to show a higher PBLH bias when the configuration includes YSU as PBL scheme, ensemble mean biases are larger in the West because RUC LSM produce higher positive bias compare to Thermal Diffusion LSM. Also, we showed in Figure 14 how these two LSMs tends to overestimate the sensible heat. These results suggest that the different LSMs could misrepresent the surface energy budget in spatially coherent ways over the region, causing spatially

coherent biases in the PBLH. Cumulus parameterization also played an important role in the PBLH, as the ensemble members that did not include a cumulus parametrization produce high positive biases. This could be explained by the lack of parameterized subgrid-scale convection that could, in reality, limit PBL growth. While we were not able to find an optimal configuration across all the sites, we did find, similar to Coniglio et al. (2013) and Milovac at el. (2016), that the MYNN PBL scheme produced the smallest PBLH biases averaged over the region.

This ensemble helped us to understand and evaluate atmospheric transport errors due to physics parameterization and reanalysis, and to understand how these transport errors are propagated into simulated PBL $CO_2$ mole fractions. However, it is important to note several limitations of this study: (1) we explore fewer microphysics and reanalysis options (only two) compared to the number of PBL, cumulus, and LSM with three options for each, (2) this evaluation was performed over a

limited period of time and location, (3) the range of parameterizations available for this sensitivity study is ad hoc and uncalibrated, and (4) the cumulus parameterizations utilized do not include parameterized transport of $CO_2$. We also note that some of the parameterizations (i.e. RUC LSM) were only run using the FNL reanalysis product, which may cause some under-estimation of the variability as this LSM contributes significantly to the errors of all the meteorological variables. We cannot yet quantify the impact of the lack of parameterized cumulus transport of $CO_2$ transport on our findings. Our

meteorological results, however, are broadly consistent with past literature. The biases found in this study are a concern, since atmospheric inversions assume atmospheric transport errors are unbiased. Since this bias exist across the different meteorological variables studied here, future selection of a least biased model may need to weight the impact of each meteorological variable on $CO_2$ or to optimize the atmospheric transport model through a data assimilation technique. A



calibrated transport ensemble may be the most efficient approach to generating an unbiased representation of atmospheric transport and associated errors.

### 5 Conclusion

In this study, we evaluated and quantified the atmospheric transport errors across a highly instrumented area, the Mid-
Continent Intensive region of the Midwest U.S., for the period June 18 to July 21 of 2008. Transport errors were quantified independently of flux errors and propagated into $CO_2$ mole fractions using a multi-physics and multi-reanalysis ensemble. Each model configuration was coupled to the same surface fluxes from CarbonTracker. We conclude that all physics parameterization except for microphysics have a significant impact on both $CO_2$ mole fractions and meteorological variables. We also found that PBLH and $CO_2$ mole fractions have similar sensitivities to the different physics schemes. The
relationship between the two variables is reinforced by the high correlations between PBLH errors and $CO_2$ mole fraction errors. Among the multiple configurations evaluated here, we intended to find the configuration best suited to represent the atmospheric transport over the region. However, we show no single model configuration was free from bias for every meteorological variable (PBLH, wind speed and wind direction) and these biases vary across the domain. Some of the physics parameterization schemes tested in this study, such as RUC LSM, YSU and MYJ PBL schemes, showed systematic
biases over the entire region, whereas the MYNN PBL scheme shows the most reasonable performance on average across the region.

The model configurations gave us additional insights into the magnitudes of the atmospheric transport errors that can be encountered over this region. However, multiple challenges remain. We showed that bias errors vary spatially across the
region. If these errors persist in the transport used for a regional inversion, these biases will be propagated into the inverse fluxes. Finally, no optimal model configuration was found for the entire region. Therefore, we conclude that both random and systematic errors will remain if any one model configuration is used. An ensemble approach, possibly combined with data assimilation, could better minimize biases and characterize the spatio-temporal structures of the atmospheric transport errors for future regional inversion system.

*Code availability*. The code is accessible under request by contacting the corresponding author (lzd120@psu.edu).

*Data availability*. Meteorological data were obtained from the University of Wyoming's online data archive (http://weather.uwyo.edu/upperair/sounding.html) for the 14 rawinsonde stations. Tower Atmospheric CO2
Concentration data set is available on-line [http://daac.ornl.gov] from Oak Ridge National Laboratory Distributed Active Archive Center, Oak Ridge, Tennessee, USA. http://dx.doi.org/10.3334/ORNLDAAC/1202. The other two towers (Park Falls-WLEF and West Branch-WBI) are part of the Earth System Research Laboratory/Global Monitoring Division





(ESRL/GMD) tall tower network (Andrews et al., 2014; https://www.esrl.noaa.gov/gmd/ccgg/insitu/). The WRF model results are accessible under request by contacting the corresponding author (lzd120@psu.edu).

*Author contribution.* L. Díaz Isaac performed the model simulations and the model-data analysis. T. Lauvaux provided
guidance with model simulations. T. Lauvaux and K. J. Davis provided guidance with the model-data analysis. All authors contributed to the design of the study and the preparation the paper.

*Competing interests.* The authors declare that they have no conflict of interest.

### Acknowledgements

This research was supported by NASA's Terrestrial Ecosystem and Carbon Cycle Program, grant NNX14AJ17G, NASA's Earth System Science Pathfinder Program Office, Earth Venture Suborbital Program, grant NNX15AG76, NASA Carbon Monitoring System, grant NNX13AP34G, and an Alfred P. Sloan Graduate Fellowship. This document benefitted from the comments of Dr. Natasha Miles, Dr. Chris E. Forest and Dr. Andrew Carleton. Data were provided by University of Wyoming's online meteorological data archive of NOAA NWS rawinsondes, NOAA's Earth System Research Laboratory
Global Monitoring Division tall tower network, the AmeriFlux network, and Penn State's contributions to the NACP Midcontinent Intensive regional study.

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




**Table 1. Different model configurations used in this study.**

| Model Number | Reanalysis | LSM Scheme | PBL Scheme | Cumulus Scheme | Microphysics Schemes |
|---|---|---|---|---|---|
| 1 | NARR | Noah | YSU | Kain-Fritsch | WSM 5-class |
| 2 | NARR | Noah | MYJ | Kain-Fritsch | WSM 5-class |
| 3 | NARR | Noah | MYNN | Kain-Fritsch | WSM 5-class |
| 4 | FNL | RUC | YSU | Kain-Fritsch | WSM 5-class |
| 5 | FNL | RUC | MYJ | Kain-Fritsch | WSM 5-class |
| 6 | FNL | RUC | MYNN | Kain-Fritsch | WSM 5-class |
| 7 | NARR | Thermal Dif. | YSU | Kain-Fritsch | WSM 5-class |
| 8 | NARR | Thermal Dif. | MYJ | Kain-Fritsch | WSM 5-class |
| 9 | NARR | Thermal Dif. | MYNN | Kain-Fritsch | WSM 5-class |
| 10 | NARR | Noah | YSU | Grell-3D | WSM 5-class |
| 11 | NARR | Noah | MYJ | Grell-3D | WSM 5-class |
| 12 | NARR | Noah | MYNN | Grell-3D | WSM 5-class |
| 13 | FNL | RUC | YSU | Grell-3D | WSM 5-class |
| 14 | FNL | RUC | MYJ | Grell-3D | WSM 5-class |
| 15 | FNL | RUC | MYNN | Grell-3D | WSM 5-class |
| 16 | NARR | Thermal Dif. | YSU | Grell-3D | WSM 5-class |
| 17 | NARR | Thermal Dif. | MYJ | Grell-3D | WSM 5-class |
| 18 | NARR | Thermal Dif. | MYNN | Grell-3D | WSM 5-class |
| 19 | NARR | Noah | YSU | Kain-Fritsch | Thompson |
| 20 | NARR | Noah | MYJ | Kain-Fritsch | Thompson |
| 21 | NARR | Noah | MYNN | Kain-Fritsch | Thompson |
| 22 | FNL | RUC | YSU | Kain-Fritsch | Thompson |
| 23 | FNL | RUC | MYJ | Kain-Fritsch | Thompson |
| 24 | FNL | RUC | MYNN | Kain-Fritsch | Thompson |
| 25 | NARR | Thermal Dif. | YSU | Kain-Fritsch | Thompson |
| 26 | NARR | Thermal Dif. | MYJ | Kain-Fritsch | Thompson |
| 27 | NARR | Thermal Dif. | MYNN | Kain-Fritsch | Thompson |
| 28 | NARR | Noah | YSU | Grell-3D | Thompson |
| 29 | NARR | Noah | MYJ | Grell-3D | Thompson |
| 30 | NARR | Noah | MYNN | Grell-3D | Thompson |
| 31 | NARR | Noah | YSU | No CP | WSM 5-class |
| 32 | NARR | Noah | MYJ | No CP | WSM 5-class |
| 33 | NARR | Noah | MYNN | No CP | WSM 5-class |
| 34 | FNL | RUC | YSU | No CP | WSM 5-class |
| 35 | FNL | RUC | MYJ | No CP | WSM 5-class |
| 36 | FNL | RUC | MYNN | No CP | WSM 5-class |
| 37 | NARR | Thermal Dif. | YSU | No CP | WSM 5-class |
| 38 | NARR | Thermal Dif. | MYJ | No CP | WSM 5-class |
| 39 | NARR | Thermal Dif. | MYNN | No CP | WSM 5-class |
| 40 | FNL | Noah | YSU | Kain-Fritsch | WSM 5-class |
| 41 | FNL | Noah | MYJ | Kain-Fritsch | WSM 5-class |
| 42 | FNL | Noah | MYNN | Kain-Fritsch | WSM 5-class |
| 43 | FNL | Thermal Dif. | YSU | Kain-Fritsch | WSM 5-class |
| 44 | FNL | Thermal Dif. | MYJ | Kain-Fritsch | WSM 5-class |
| 45 | FNL | Thermal Dif. | MYNN | Kain-Fritsch | WSM 5-class |



**Table 2. WRF physical parameterizations included in the sensitivity analysis.**

| Parameter | Options |
|---|---|
| **Land Surface Model** | Noah (Chen and Dudhia, 2001)<br>Rapid Update Cycle (RUC; Smirnova, 2000)<br>5-layer Thermal Diffusion (Dudhia, 1996) |
| **Planetary Boundary Layer (PBL) scheme** | Yonsei University (YSU; Hong et al., 2006)<br>Mellor-Yamada-Janjic (MYJ; Janjic, 2002)<br>Mellor-Yamada-Nakanishi-Niino Level 2.5 (MYNN2.5; Nakanishi & Niino, 2004) |
| **Surface Layer** | MM5 similarity/YSU PBL scheme<br>Eta Similarity/MYJ PBL scheme<br>MYNN surface layer/MYNN PBL scheme |
| **Cumulus** | Kain-Fritsch (KF; Kain, 2004)<br>Grell-3Devenyi (G3D; Grell and Devenyi, 2002)<br>No cumulus parameterization |
| **Microphysics** | WSM 5-class (Hong et al., 2004)<br>Thompson et al., (2004) |
| **Shortwave/Longwave radiation physics** | Dudhia/Rapid Radiative Transfer Model (RRTM) |
| **Initial & Boundary Conditions** | North America Regional Reanalysis (NARR)<br>Global Final Analysis (FNL) |





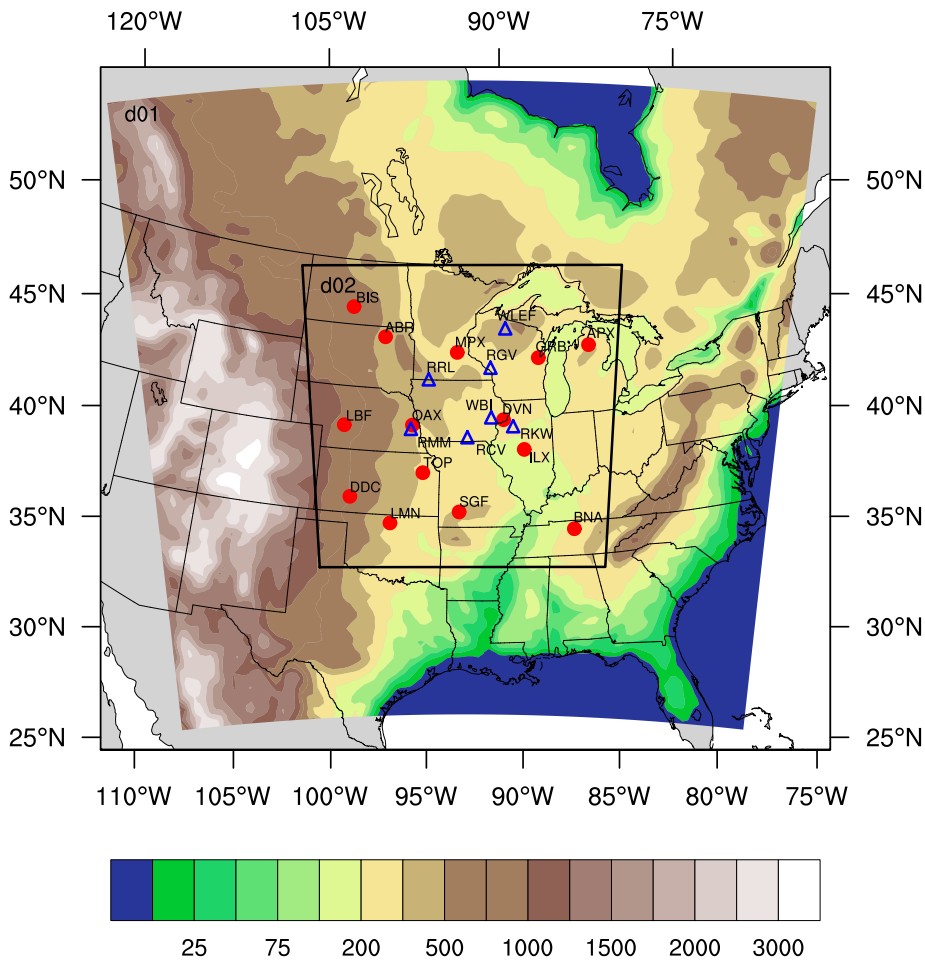

**Figure 1: Geographical domain used by the WRF-ChemCO$_2$ physics ensemble. The parent domain (d01) is resolved at 30-km in the horizontal, the inner domain (d02) at 10-km. The color shading represents modeled terrain height in meters above sea level. The inner domain covers the study region and includes the rawinsonde sites (red circles) and the CO$_2$ towers (blue triangles) locations.**




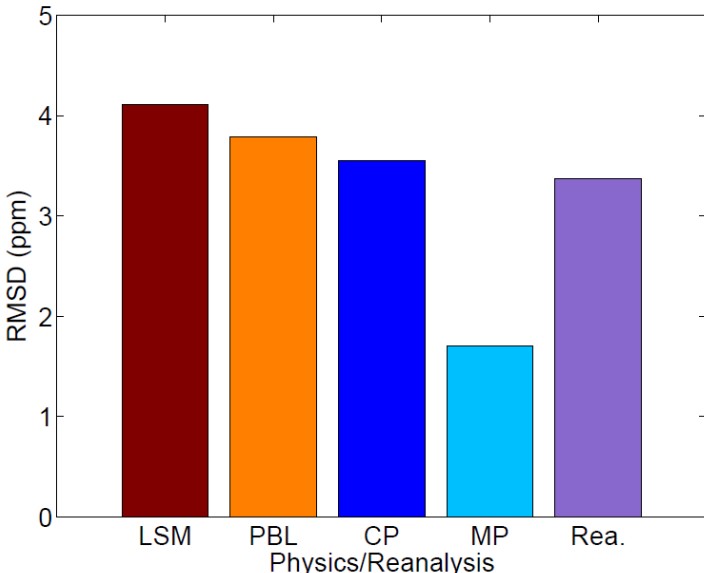

**Figure 2. Sensitivity of $CO_2$ mole fractions as a function of model physics parameterizations (i.e., land surface model (LSM), planetary boundary layer scheme (PBL), cumulus parameterization (CP), microphysics parameterization (MP) and Reanalyses). The root mean square difference (RMSD) of the $CO_2$ mole fractions simulated at each site and for each model ensemble member was computed by varying only the type of physics parameterization noted, and keeping all other model elements constant. RMSD was averaged across sites and across model ensembles.**





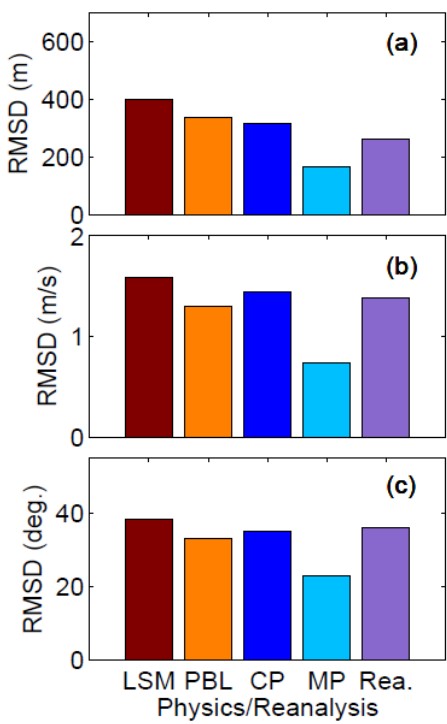

**Figure 3. Root mean square difference (RMSD) of the PBLH (a), wind speed (b) and wind direction (c) for the different physics parameterizations (i.e., land surface model (LSM), planetary boundary layer scheme (PBL), cumulus parameterization (CP), microphysics parameterization (MP) and Reanalyses). The RMSDs were computed in the same way for these variables as for PBL**
5 **$CO_2$ in Figure 2.**



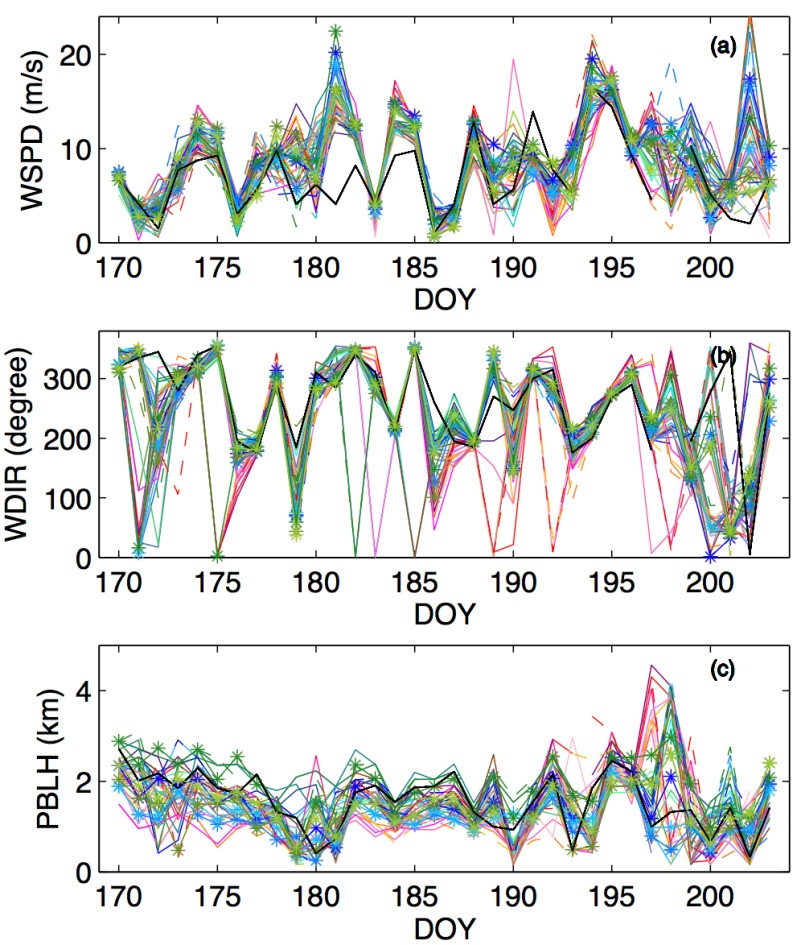

**Figure 4. Observed (black line) and simulated (colored lines, see Table 1) PBL (300 m AGL) wind speed (a), wind direction (b) and PBLH (c) at time 0000 UTC from day of the year (DOY) 169 to 203 of 2008 at the Chanhassen, Minneapolis (MPX) rawinsonde site.**


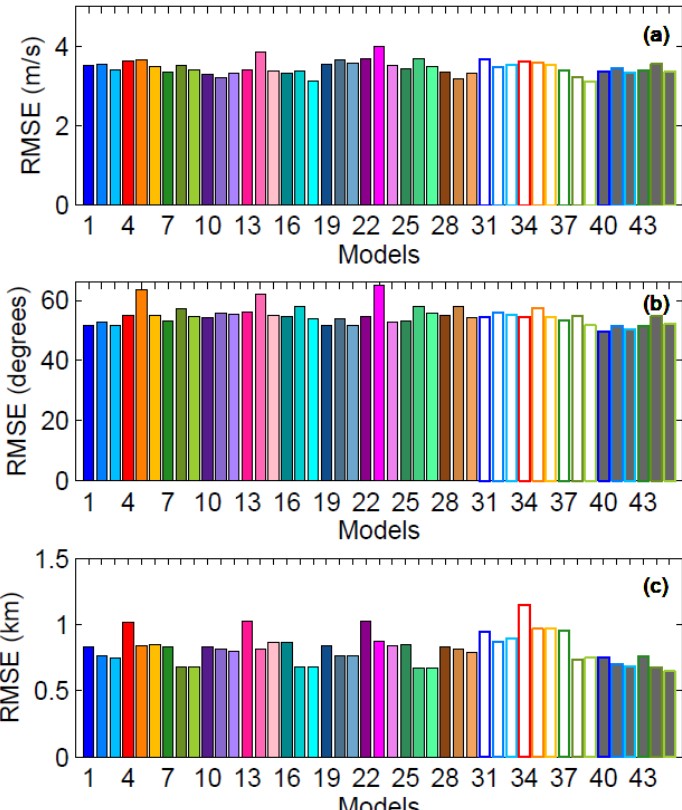

**Figure 5. Regional averages of the monthly average of wind speed (a), wind direction (b) and PBLH (c) RMSE for the different models (see Table 1 for model configurations).**



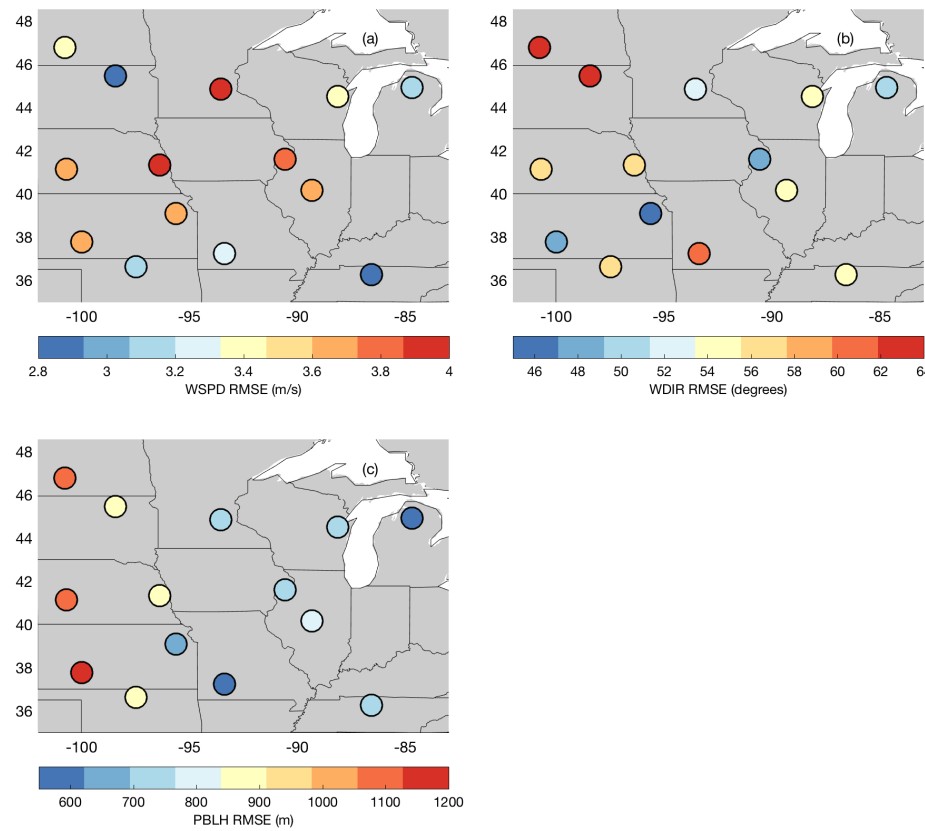

**Figure 6. Ensemble mean of monthly averaged RMSE of wind speed (a), wind direction (b) and PBLH (c).**





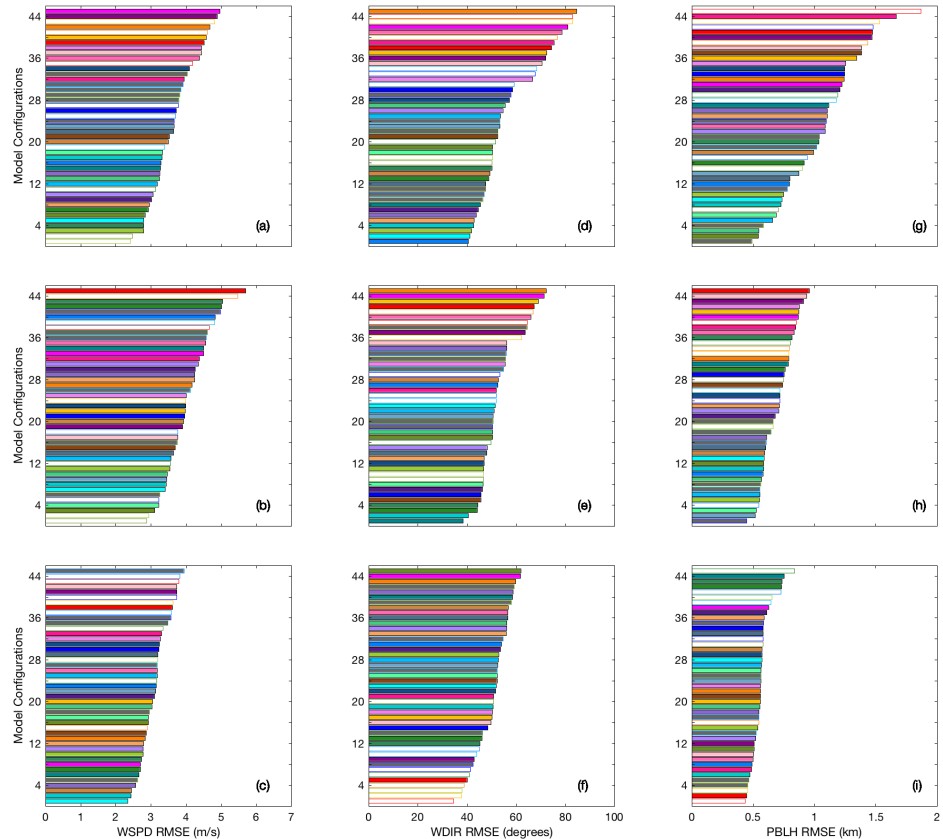

**Figure 7. Monthly average wind speed (a-c), wind direction (d-f) and PBLH (g-i) RMSE for rawinsonde sites LBF (first row), MPX (second row) and APX (third row). Models are sorted from the smallest to the highest RMSE. Model configurations are ordered by RMSE and identified by color (see Table 1).**



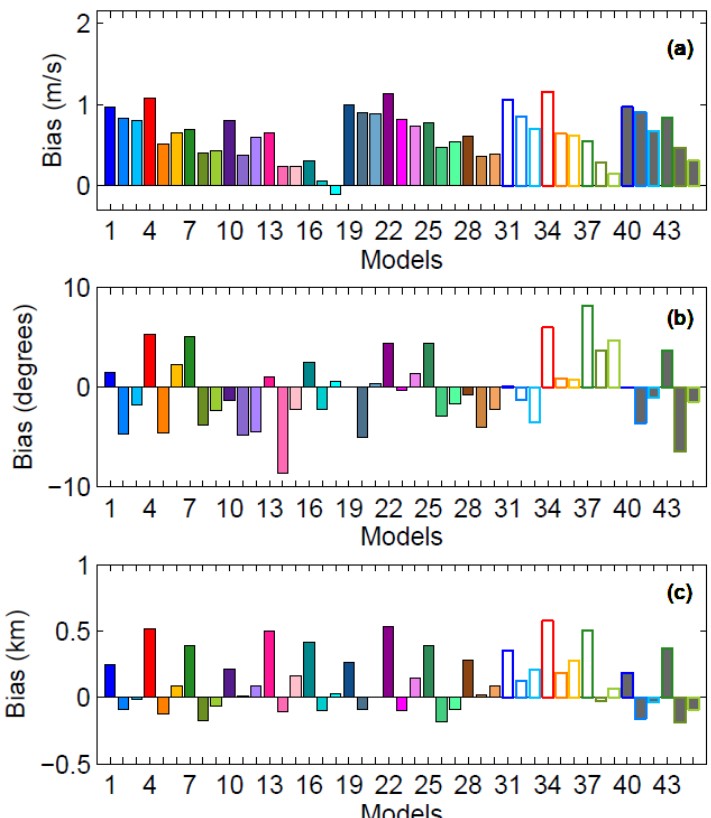

**Figure 8. Regional average of the monthly average of PBL wind speed (a), PBL wind direction (b) and PBLH (c) bias for the different model configurations (identified by number and color - see Table 1).**





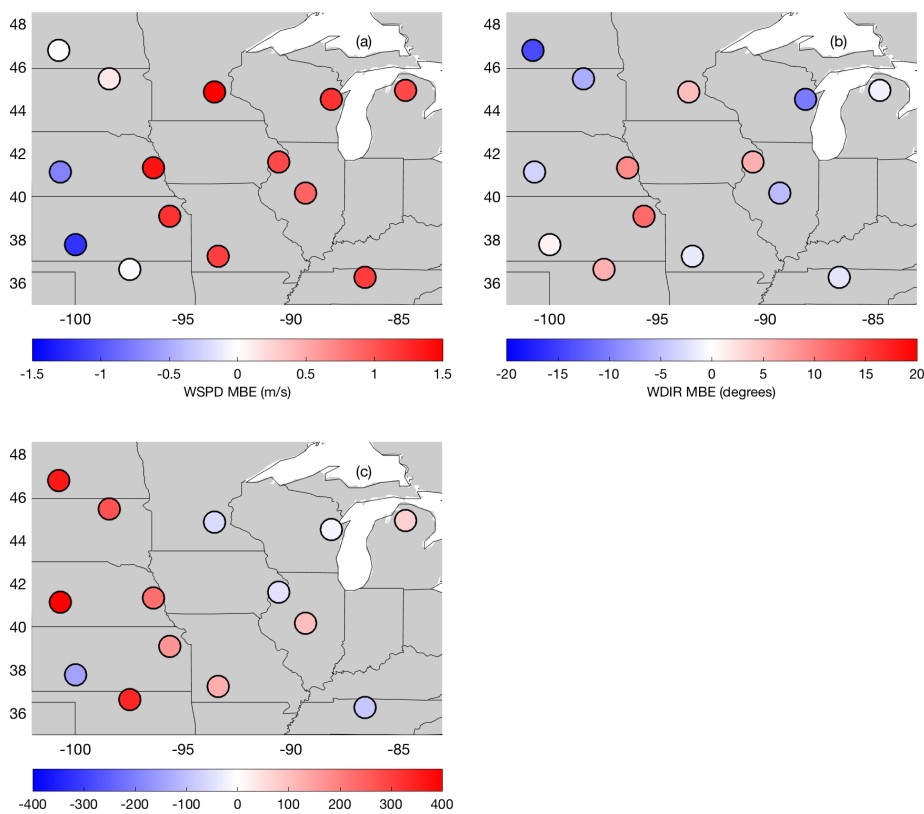

**Figure 9.** Ensemble mean of the mean bias error (MBE) for PBL wind speed (a), PBL wind direction (b) and PBLH (c).




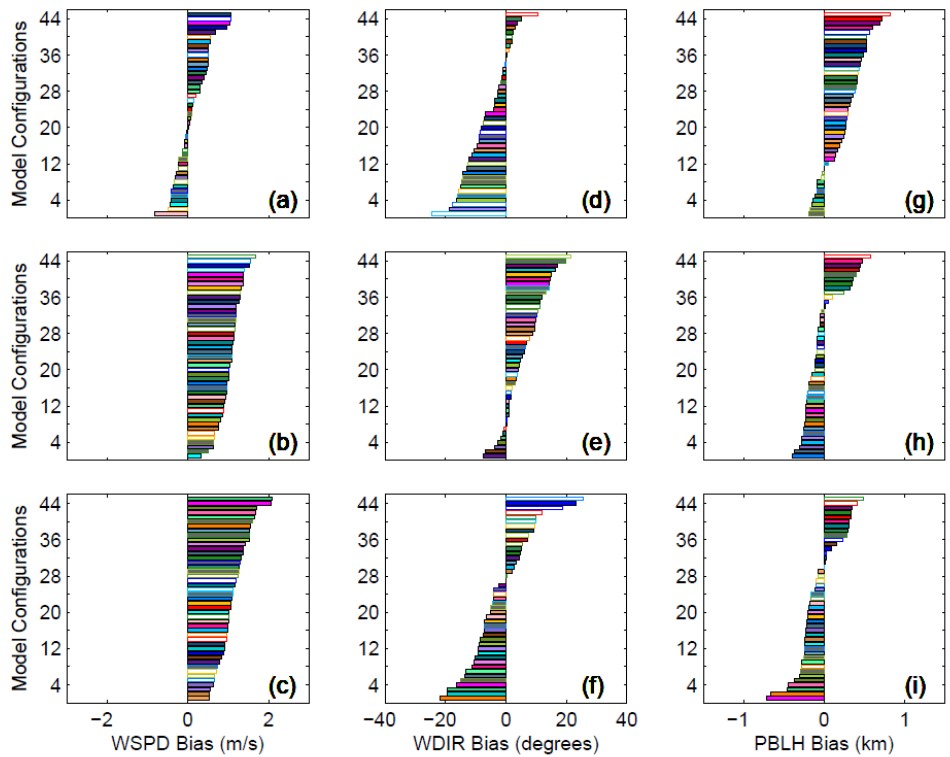

**Figure 10. Monthly average wind speed (a-c), wind direction (d-f) and PBLH (g-i) MBE for rawinsonde sites ABR (first row), DVN (second row) and BNA (third row). Models are sorted from the negative to the positive bias. Model configurations are ordered by MBE and identified by color (see Table 1).**

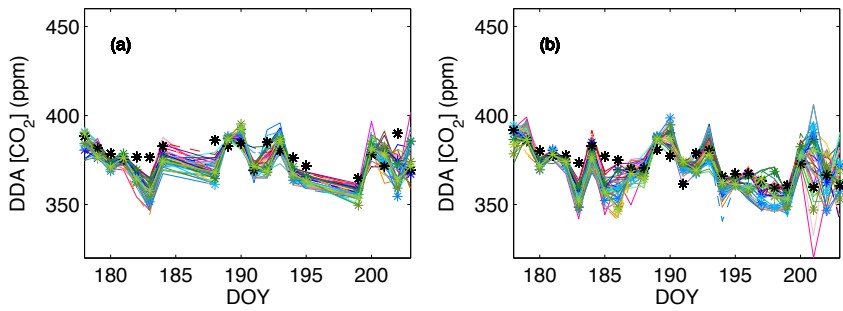

**Figure 11. Observed (black stars) and simulated (colored lines) DDA $CO_2$ mole fraction (ppm) at Centerville (RCV) (a) and WBI (b).**





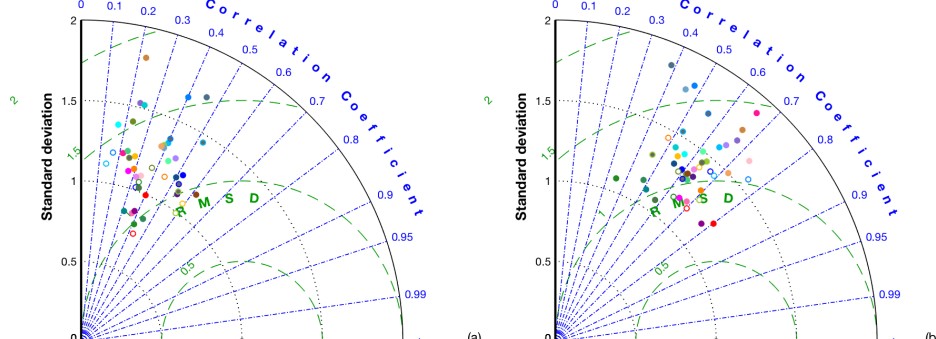

**Figure 12. Taylor Diagram comparing observations versus simulations at (a) Mead (RMM) and (b) West Branch (WBI), using DDA $CO_2$ mole fractions from 100 m AGL. Black dots at (1, 1) represent the observations.**





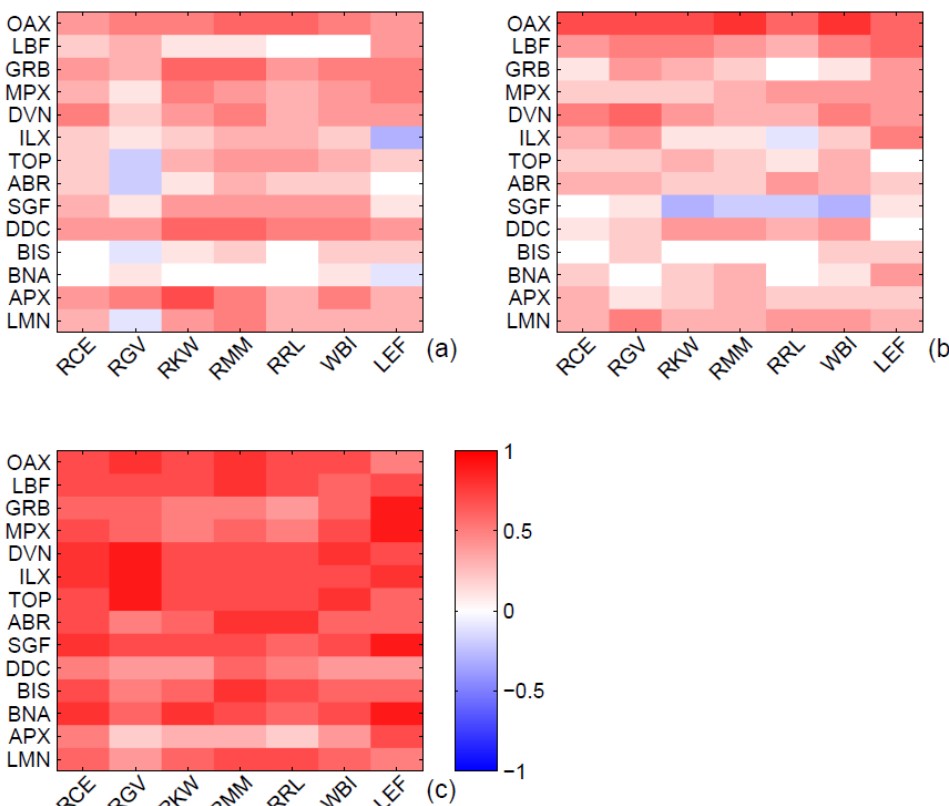

**Figure 13. Tower and rawinsonde site specific spatial correlation coefficients between ensemble mean MBE of (a) wind speed, (b) wind direction and (c) PBLH and ensemble mean MBE of DDA $CO_2$ mole fractions. The abscissa shows the different $CO_2$ tower sites, while the ordinate shows rawinsonde sites.**





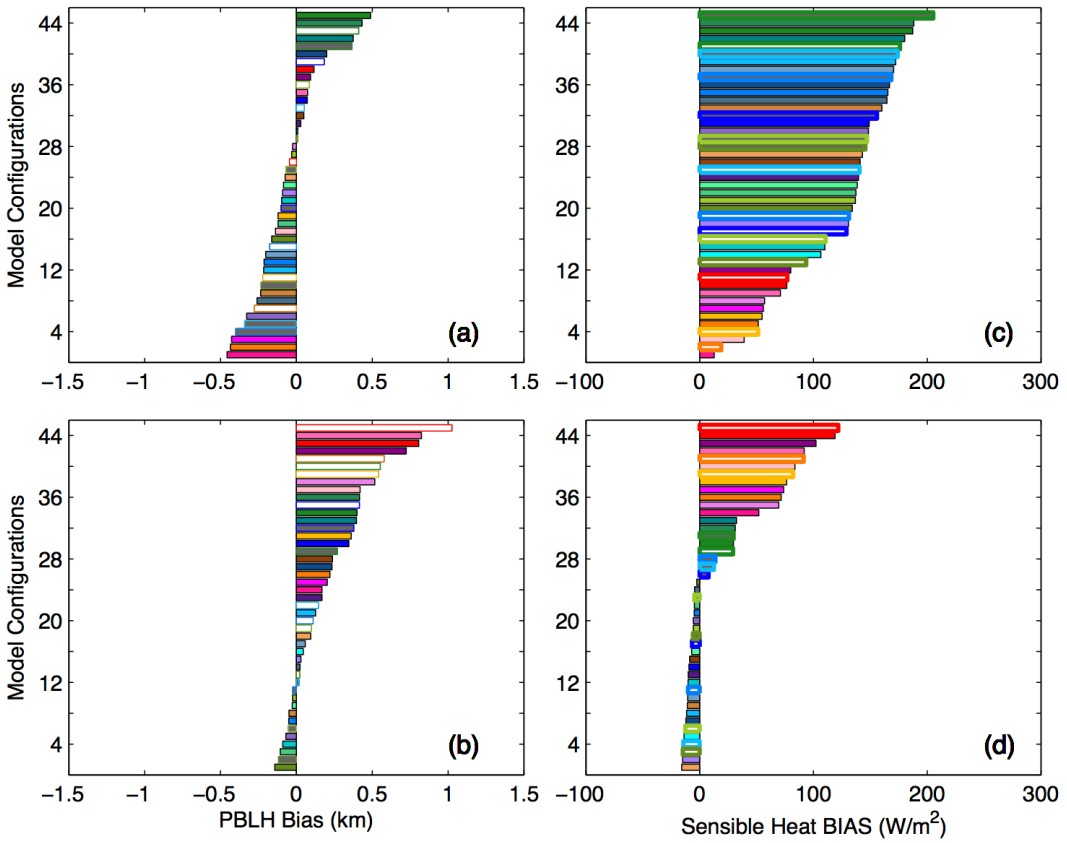

**Figure 14. Monthly averaged PBLH MBE for rawinsonde sites (a-b) and sensible heat MBE for eddy covariance sites (c-d). The two rawinsode sites MPX (a) and OAX (b) are close to the eddy covariance sites USKUT (c) and USNe3 (d), respectively. Model configurations are identified by color (see Table 1).**

