# Peer review of "Impact of physical parameterizations and initial conditions on simulated atmospheric transport and $CO_2$ mole fractions in the US Midwest"

_Atmospheric Chemistry and Physics, 2018_

## Short Comment (SC1) · 19 Mar 2018

Please revise text to cite CT2009 following the instructions at https://www.esrl.noaa.gov/gmd/ccgg/carbontracker/citation.php.

---

## Referee Comment (RC1) · Anonymous Referee #2 · 14 May 2018

This paper presents the evaluation of multiple WRF model configurations over the US Upper Midwest. The model configurations are constructed by selecting different PBL, cumulus and microphysics schemes. These configurations are also used to simulate atmospheric CO2 mole fractions by using the CO2 fluxes from the Carbon Tracker global inversion system. It is important to use well constrained meteorological drivers in CO2 inversion studies. As the previous studies have shown, uncertainties in the meteorological drivers can lead to large discrepancies in simulations of atmospheric CO2 mole fractions.

[Figure]

This study is trying to identify how much different physics parameterizations contribute to error in CO2 transport within the WRF-CO2 model. I have some reservations on the setup and conduct of the modeling exercises and evaluations in the paper:

The WRF-Chem CO2 model needs more explanation. How different is this model from the regular WRF-Chem model, which can simulate passive tracers including CO2? How vertical mixing of CO2 is parameterized in this model configuration? Does this version of the model include convective tracer transport?

The spatial resolution of the inner domain (10km) isn't very high. Given the importance of the representation of heterogeneity in simulation of both anthropogenic and biospheric CO2 fluxes, it'd greatly help to conduct higher resolution simulations (2-4 km).

The vertical resolution of the domain is quite high (40 levels <2km) near surface. However, there are only 19 levels above 2km. How the vertical levels aloft are distributed? The vertical grid spacing will impact how the model captures capping inversion layer, clouds and so on.

14 radiosounding sites are used to evaluate the meteorological simulations. The accurate simulation of vertical mixing is important for CO2 simulations. Unfortunately, the regular radiousoundings in the US don't capture (00 and 12Z only) the deep boundary layers during the afternoon hours in the Midwest due to timing. I recommend using additional data (e.g. ceilometers) to evaluate the daytime evolution of the boundary layers in Midwest.

The choice of the meteorological data for the model verifications is limited to the radiosounding data, which are quite limited in time and space. I suggest adding surface wind, temperature and other measurements to the model evaluations.

It's recommended to use the GF cumulus paramaterization instead of the G3 scheme in WRF. I suggest testing the model with the GF scheme.

There is one key uncertainty associated with using the CO2 fluxes from CT in WRF-CO2 modeling here. The spatial resolution of the EDGAR and CASA CO2 emissions are much coarser than the inner WRF grid spacing here. Consequently, the WRF-CO2 model can't capture the regional CO2 variability even with "perfect" meteorology. This needs to be discussed in this paper. At least, high resolution anthropogenic CO2 emissions (e.g. VULCAN) could be used in such model setup.

Minor comments: Abstract: "...is this variability is..."?

For the WRF model, please cite the recent WRF/WRF-Chem description paper: Powers et al., Weather Research and Forecasting Model: Overview, System Efforts, and Future Directions, AMS. https://doi.org/10.1175/BAMS-D-15-00308.1

––––––––––––––––––––––––––––––––

---

## Referee Comment (RC2) · Anonymous Referee #1 · 10 Jun 2018

Diaz-Isaac et al. studied the impact of transport errors on simulated CO2 mole fractions in the US Midwest, which is very relevant to the goal of improving the estimate of surface CO2 fluxes, since transport models are used to derive surface CO2 fluxes in an inverse analysis. The authors tested a series of physical parameterizations and initial conditions and pointed out that most tested physical parameters and initial conditions have a significant impact on simulated CO2, either influencing the planetary boundary layer height (PBLH) that confirms the previous finding that the correct representation of PBLH is important for accurate CO2 simulations or changing wind speed and direction.

[Figure]
* * *
Interactive
comment

The paper is well structured and clearly written. The reviewer suggests publication after the following concerns have been addressed.

One of the major conclusions that "all physics parameterization except for microphysics have a significant impact on both CO2 mole fraction sand meteorological variables" is based on the magnitude of the simulated CO2 root mean square difference (RMSD). The authors mentioned that it was computed for each model ensemble member by varying only the type of physics parameterization. My understanding is that for the LSM scheme, multiple sets of ensemble members can be used for the computation, e.g. models nos. 1&7, nos. 2&8, nos. 3&9, nos. 22&40&43, nos.23&41&44, nos.24&42&45. Was the presented RMSD for LSM the mean of all different sets of ensemble members? For the calculation of the mean of the ensemble for day i in equation 1, are all 45 ensembles used or only the sets with varying one type of physics parameterization? This should be clarified.

I agree with the concerns raised by reviewer #2 on the use of the 14 radiosounding sites. The vertical profiles of temperature and CO2 mole fractions at multiple sites from the NOAA aircraft program and/or from other intensive campaigns could be looked into, at least for the PBLH.

P5/L6: use longwave instead of long wave

P12/L22, P13/L5, P15/L21: use the plural form "show" instead of "shows"

L15/22: "We did not find any relationship between error correlation and distance". It would be convincing to show a scatter plot, error correlation vs. distance for each grid in Figure 13, or at least present the correlation of error correlation vs. distance.

L16/11-12: provide evidence of PBL winds impacting the distribution and magnitude of the inverse CO2 fluxes over the region.

The reviewer found it difficult to obtain any meaningful information from the figures 7,10&14 where the results of all 45 model results are presented for three selected

sites. The results should be first summarized before being presented, or simply be moved to the supplementary.

---

## Author Comment (AC1) · 27 Jul 2018

Thanks for letting us know how to cite appropriately CarbonTracker. We changes the citations as suggested:

P7/L11-L12: "For this study, we used the summer 2008 posterior surface fluxes from the data assimilation system CarbonTracker version CT2009 (Peters et al., 2007; with updates documented at http://carbontracker.noaa.gov)."

P19/L26-L27: "CarbonTracker CT2009 results provided by NOAA ESRL, Boulder, Col-

orado, USA from the website at http://carbontracker.noaa.gov."

---

## Author Comment (AC2) · 27 Jul 2018

**Answers to Referee #2 comments:** *Review of Impact of physical parametrizations and initial conditions on simulated atmospheric transport and $CO_2$ mole fractions in the US Midwest*

We thank the referee for the helpful comments that will improve the manuscript. In the text below, we have tried our best to respond to all the general and specific comments provided by the reviewer.

**Comments to Author:**

This paper presents the evaluation of multiple WRF model configurations over the US Upper Midwest. The model configurations are constructed by selecting different PBL, cumulus and microphysics schemes. These configurations are also used to simulate atmospheric $CO_2$ mole fractions by using the $CO_2$ fluxes from the Carbon Tracker global inversion system. It is important to use well constrained meteorological drivers in $CO_2$ inversion studies. As the previous studies have shown, uncertainties in the meteorological drivers can lead to large discrepancies in simulations of atmospheric $CO_2$ mole fractions.

This study is trying to identify how much different physics parameterizations contribute to error in CO2 transport within the WRF-$CO_2$ model. I have some reservations on the setup and conduct of the modeling exercises and evaluations in the paper:

**REF-C1:** The WRF-Chem $CO_2$ model needs more explanation. How different is this model from the regular WRF-Chem model, which can simulate passive tracers including $CO_2$?
*Author-C1*: The WRF-Chem $CO_2$ module has been developed following the passive tracer scheme of WRF-Chem, modified to include multiple tracers (from 10 to 25 tracers currently). The physics is identical to the passive tracer original module, using the turbulent mixing in the Planetary Boundary Layer and the mean wind fields for advection and diffusion elsewhere.

**REF-C2:** How vertical mixing of $CO_2$ is parameterized in this model configuration? Does this version of the model include convective tracer transport?
*Author-C2*: In the WRF-Chem version used in this paper, the only convective parameterization available is based on precipitation rates to diagnose the vertical mass flux (1D) with a pre-defined cloud base and cloud top. We are currently developing the coupling between 3D mass fluxes from the Kain-Fritsch scheme with the chemistry $CO_2$ module in order to properly account for convective transport. This work is still ongoing. In this paper, we used a resolution of 10km that may under-estimate convective transport for small-scale features but should be sufficient to resolve some of the large-scale convective events in summer. We added the following sentences to clarify in the text in ***bold/italic***:
P17/L31-L34: ***The impact of CP on $CO_2$ mole fractions requires more evaluation, because our convective scheme is not coupled with the tracers (i.e., $CO_2$ mole fractions), however, we can still use the convective schemes to evaluate its impact on wind fields and PBLH.*** Consequently, we cannot yet quantify the impact of the lack of parameterized cumulus transport of $CO_2$ transport on our findings.

**REF-C3:** The spatial resolution of the inner domain (10km) isn't very high. Given the importance of the representation of heterogeneity in simulation of both anthropogenic and biospheric $CO_2$ fluxes, it'd greatly help to conduct higher resolution simulations (2-4 km).

*Author-C3*: Certainly, having a higher horizontal resolution model (i.e., 2-4 km) will improve at least the atmospheric transport over the US Upper Midwest region and provide more variability or heterogeneity to simulated $CO_2$ mole fractions linked with weather systems. However, in terms of $CO_2$ mixing ratios during non-frontal conditions, the spatial scale of structures over the US Upper Midwest is on the order of tens of kilometers. Therefore, increasing the spatial resolution is not expected to improve the representativity of the simulation results. High-resolution simulations would, however, increase considerably the required computational time to execute multiple simulations over a large domain (i.e. 1,600x1,600km). We also note that WRF-Chem is coupled to the only operational inversion product available for $CO_2$ fluxes at 1°×1°, CarbonTracker. If higher-resolution fluxes become available, the WRF grid spacing could be increased to explore fine-scale structures.

We add a note about the decision of the 10-km resolution in the text (***bold/italic***):
P4/L24-26: The coarse domain (d01) uses a horizontal grid spacing of 30-km and the nested or inner domain (d02) uses 10-km grid spacing (Figure 1). ***Because of limited computational time and the resolution of the $CO_2$ surface fluxes described on section 2.6, we decided to keep our highest resolution of the model up to 10-km.***

**REF-C4:** The vertical resolution of the domain is quite high (40 levels <2km) near surface. However, there are only 19 levels above 2km. How the vertical levels aloft are distributed? The vertical grid spacing will impact how the model captures capping inversion layer, clouds and so on.

*Author-C4*: This was a mistake in the document. We have a total of 59 vertical levels where 40 levels are within the first 4km of the atmosphere instead of the first 2km (see Figure1). Figure 1, shows a vertical profile of temperature at OAX site for configuration of our ensemble. The first black line shows the first 40 levels, above level 40 the figure includes dashed lines every 5 levels up to level 59. Because the statement in the article was incorrect, we end up fixing line 22-23 on page 4 as follow:
P4/L21-L22: "*The atmospheric column in each simulation is described with 59 vertical levels, with 40 of them within the first 4-km of the atmosphere. Two nested domains were used.*"

[Figure]

**Figure 1.** Vertical profile of potential temperature at OAX rawinsonde site from Model 1 on DOY 175 at 0000 UTC. Black solid line represents the height at level 40 of the model and the dashed lines shows the height of every 5 level.

**REF-C5:** 14 radiosounding sites are used to evaluate the meteorological simulations. The accurate simulation of vertical mixing is important for $CO_2$ simulations. Unfortunately, the regular radiosoundings in the US don't capture (00 and 12Z only) the deep boundary layers during the afternoon hours in the Midwest due to timing. I recommend using additional data (e.g. ceilometers) to evaluate the daytime evolution of the boundary layers in Midwest.

*Author-C5*: We agree with the reviewer that radionsonde data are collected at the same time of day, usually after the PBL height peak, and cover only certain parts of the domain. We considered adding more observation to our evaluation but decided not to for several reasons. The radiosonde data provides the best spatial and temporal resolution that we can have for our project. Other data such as $CO_2$ mole fractions from the NOAA aircraft program (or other aircraft campaigns) will bring the limitation of the time and/or spatial coverage. Only five sites are within our WRF simulation domain and profiles are usually collected every two weeks (i.e. about 2 per site over our simulation period). Most aircraft campaigns do not sample the mixing depth but rather collect long transects over the continent (e.g. COBRA, ATom). In addition, we have no available intensive campaign in that part of the country and for the simulation period we selected. In the near future, the Atmospheric Carbon & Transport (ACT) project funded by NASA, that is currently performing multiple fields campaigns over the East and Midwest of United States for different seasons, will provide a significant spatial coverage to address transport model errors. The campaign has just finished its fourth deployment in the Spring of 2018. Regarding surface stations, including them in our analysis would bias our results towards near-surface model errors, which we know are not representative of the whole Planetary Boundary Layer (PBL) (e.g. Hu et al., 2010; Rogers et al., 2013; Deng et al., 2017). $CO_2$ molecules are mixed over the entire PBL during daytime. Therefore, adding tens to hundreds of surface stations will not represent the actual model errors in the PBL. For these different reasons, we decided to focus on operational radiosondes launched at 00z, using

mid-PBL wind measurements. Because we are aware of this limitation in our study and this is a concerned from both reviewers we decided to add text to the last paragraph of our Discussion section (P18/L1-L3).

**Citation:** Rogers, RE, Deng, A, Stauffer, DR, Gaudet, BJ, Jia, Y, Soong, S, Tanrikulu, S, 2013, Application of the Weather Research and Forecasting Model for Air Quality Modeling in the San Francisco Bay Area. *J. Appl. Meteor.*, 52: 1953–1973. DOI: https://doi. org/10.1175/JAMC-D-12-0280.1

**REF-C6:** The choice of the meteorological data for the model verifications is limited to the radiosounding data, which are quite limited in time and space. I suggest adding surface wind, temperature and other measurements to the model evaluations.
*Author-C6*: We are aware of the limited data used to explore the transport errors. However, our main interest as mentioned on section 2.8 of the paper is to evaluate meteorological observations at height similar to the observation that means that we want to use observations that are within the well-mixed layer. Surface station will fall out of the height of our interest and can represent more the surface layer than the convective layer. With respect to the variables explored, we also clarified on Section 2.9, that our main interest was to explore all the variables that we assumed contribute the most to the representations of the $CO_2$ mole fraction distributions within the PBL. Additional to that, current atmospheric inversions do not use surface wind speed and direction as input, however, it uses these two variables within the PBL.

It will be ideal a study over a region and a period of time where multiple observations can be used to have broader view of the transport uncertainty. This is one of the main goals of the Atmospheric Carbon & Transport (ACT) project by NASA, that is currently performing multiple fields campaigns over the East and Midwest of United States for different seasons. ACT main objective is to understand how different weather systems transport different greenhouse gases (GHGs), evaluate this transport and improve the estimates of sources and sinks of these GHGs.
Because we are aware of this limitation in our study and this is a concerned from both reviewers we decided to add this to the last paragraph of our Discussion section, where we introduce the different limitation of this study (see ***bold/italic*** lines in the next paragraph).

P17/L1-L3: "***We also note that models were compared only to rawinsonde data, the only type of observation that had both the temporal and vertical resolution needed to evaluate the models within the PBL. More observations with higher temporal, spatial and vertical resolution will be an asset for future evaluation of transport models, focusing on intensive campaigns over multiple seasons.*** Our meteorological results, however, are broadly consistent with past literature."

**REF-C7:** It's recommended to use the GF cumulus parameterization instead of the G3 scheme in WRF. I suggest testing the model with the GF scheme.
*Author-C7*: We have selected the Grell-3D and Kain-Fritsch schemes to represent two families of convective parameterization in order to capture model sensitivities. Recent papers suggested the Grell-Freitas parameterization to produce more reliable results in WRF simulations. We selected the older version which has been used extensively in the literature when we designed this study. While modelers often aim at simulating the atmospheric dynamics the most accurately, we focus here on the differences when using various combinations of schemes, and found that surface

schemes and PBL schemes are the most critical parameterizations. The convective parameterizations have also some impact on our results to a lower extent, but no combination of surface and PBL was able to capture the variability in mixing ratios without systematic errors. We concluded that no single configuration can guarantee a reliable representation of near-surface $CO_2$ transport, and recommended to use ensemble-based approaches. Despite the use of the GF scheme, the biases from PBL and surface schemes will remain. We added the following sentence to encourage future ensemble approaches to include the GF scheme:

P17L28-L31: ***Also, as noticed in recent studies, the Grell-Freitas convection scheme produced more reliable simulations of the atmospheric dynamics (Gao et al., 2017; Gbode et al., 2018). Therefore, we recommend the use of newly developed schemes for future studies as model schemes are made available in new model versions.***

**REF-C8:** There is one key uncertainty associated with using the $CO_2$ fluxes from CT in WRF-$CO_2$ modeling here. The spatial resolution of the EDGAR and CASA $CO_2$ emissions are much coarser than the inner WRF grid spacing here. Consequently, the WRF- $CO_2$ model can't capture the regional $CO_2$ variability even with "perfect" meteorology. This needs to be discussed in this paper. At least, high resolution anthropogenic $CO_2$ emissions (e.g. VULCAN) could be used in such model setup.

*Author-C8*: In the Upper Midwest, the spatial gradients in $CO_2$ fluxes are on the order of tens of kilometers, mostly driven by land cover types. Corn and soybean fields extend over wide areas (entire Iowa, Illinois, and beyond), as well as forested land in Missouri, and wheat fields in the western part of the region. For that reason, major spatial gradients are correctly represented at 1x1 degree resolution. Currently, a high-resolution CASA simulation at 500-m resolution has been developed and will be coupled to WRF-Chem. In future studies we plan to investigate differences in $CO_2$ fluxes. The main concern comes from the location and magnitude of the sink, as CarbonTracker is a global inversion model not aimed at representing regional fluxes. The summer drawdown and seasonal cycle show differences that could impact the $CO_2$ mixing ratios. However, no significant differences are expected in terms of spatial gradient representativity.

We added the following sentence on page 14 (***bold/italic***):
P14/L30-L31: This result suggests that transport model errors from our ensemble only represent a fraction of the total uncertainty in our modelling system. ***In this study we use CarbonTrakcer fluxes which is a global inversion system and does not aim to represent regional fluxes.*** Therefore, Additional errors can be due to incorrect $CO_2$ surface fluxes and boundary conditions.

**REF-C9**: Minor comments: Abstract: ". . .is this variability is. . ."?
*Author-C9*:  Thanks for the comment, this part of the sentence was fixed as follow:

*"PBL height varies across ensemble members by 300 to 400 m, and this variability is controlled by the same physics parameterizations."*

**REF-C10:** For the WRF model, please cite the recent WRF/WRF-Chem description paper: Powers et al., Weather Research and Forecasting Model: Overview, System Efforts, and Future Directions, AMS. https://doi.org/10.1175/BAMS-D-15-00308.1

***Author-C10***: Thanks for the reference, however, the article that you are suggesting is more an overview of WRF-ARW and WRF-Chem. For this type of paper, we would prefer to cite a more technical paper such as Skamarock et al. (2005).

---

## Author Comment (AC3) · 27 Jul 2018

**Answers to Referee #1 comments:** *Review of Impact of physical parametrizations and initial conditions on simulated atmospheric transport and $CO_2$ mole fractions in the US Midwest*

We thank the referee for the helpful comments that will improve the manuscript. In the text below, we have tried our best to respond to all the general and specific comments provided by the reviewer.

**Comments to Author:**

Diaz-Isaac et al. studied the impact of transport errors on simulated $CO_2$ mole fractions in the US Midwest, which is very relevant to the goal of improving the estimate of surface $CO_2$ fluxes, since transport models are used to derive surface $CO_2$ fluxes in an inverse analysis. The authors tested a series of physical parameterizations and initial conditions and pointed out that most tested physical parameters and initial conditions have a significant impact on simulated CO2, either influencing the planetary boundary layer height (PBLH) that confirms the previous finding that the correct representation of PBLH is important for accurate $CO_2$ simulations or changing wind speed and direction.

The paper is well structured and clearly written. The reviewer suggests publication after the following concerns have been addressed.

**REF-C1:** One of the major conclusions that "all physics parameterization except for microphysics have a significant impact on both CO2 mole fractions and meteorological variables" is based on the magnitude of the simulated CO2 root mean square difference (RMSD). The authors mentioned that it was computed for each model ensemble member by varying only the type of physics parameterization. My understanding is that for the LSM scheme, multiple sets of ensemble members can be used for the computation, e.g. models nos. 1&7, nos. 2&8, nos. 3&9, nos. 22&40&43, nos.23&41&44, nos.24&42&45. Was the presented RMSD for LSM the mean of all different sets of ensemble members? For the calculation of the mean of the ensemble for day i in equation 1, are all 45 ensembles used or only the sets with varying one type of physics parameterization? This should be clarified.

*Author-C1*: Yes, to compute the RMSD we create multiple set of ensemble members and the result that we present is the mean of that different sets of ensemble members. This same process is used for the other physics parameterizations (i.e., PBL schemes, Cumulus, Microphysics) and Reanalysis data set. RMSD presented not only for the LSM, but for PBL schemes, Cumulus, Microphysics and Reanalysis was the mean of all the different sets of the ensemble members that we were able to generate. For the calculation of the mean of the ensemble for day *i* in equation 1, we only use the set of ensemble members for the different varying type of physics parameterization.

We clarify this in the methods Sections 2.9.a, as follow (see ***bold italic***):

P9/L24-L26: The RMSD was estimated for the different physics parameterization used (i.e., LSM, PBL schemes, CP, MP) and reanalysis. ***A different set of ensembles were created for each of the physics parameterization, where the model configuration remained identical except for the tested physics parameterization and the different set of members were to compute the ensemble mean.***

**REF-C2:** I agree with the concerns raised by reviewer #2 on the use of the 14 radiosounding sites. The vertical profiles of temperature and CO2 mole fractions at multiple sites from the NOAA aircraft program and/or from other intensive campaigns could be looked into, at least for the PBLH. *Author-C2:* We considered adding more observation to our evaluation but decided not to for several reasons. The radiosonde data provides the best spatial and temporal resolution that we can have for our project. Other data such as $CO_2$ mole fractions from the NOAA aircraft program (or other aircraft campaigns) will bring the limitation of the time and/or spatial coverage. Only five sites are within our WRF simulation domain and profiles are usually collected every two weeks (i.e. about 2 per site over our simulation period). Most aircraft campaigns do not sample the mixing depth but rather collect long transects over the continent (e.g. COBRA, ATom). In addition, we have no available intensive campaign in that part of the country and for the simulation period we selected. In the near future, the Atmospheric Carbon & Transport (ACT) project funded by NASA, that is currently performing multiple fields campaigns over the East and Midwest of United States for different seasons, will provide a significant spatial coverage to address transport model errors. The campaign has just finished its fourth deployment in Spring of 2018. Regarding surface stations, including them in our analysis would bias our results towards near-surface model errors, which we know are not representative of the whole Planetary Boundary Layer (PBL) (e.g. Hu et al., 2010; Rogers et al., 2013; Deng et al., 2017). $CO_2$ molecules are mixed over the entire PBL during daytime. Therefore, adding tens to hundreds of surface stations will not represent the actual model errors in the PBL. For these different reasons, we decided to focus on operational radiosondes launched at 00z, using mid-PBL wind measurements. Because we are aware of this limitation in our study and this is a concerned from both reviewers we decided to add this to the last paragraph of our Discussion section, where we introduce the different limitation of this study (see ***bold/italic*** lines in the next paragraph).

P18/L1-L3: "***We also note that models were compared only to rawinsonde data, the only type of observation that had both the temporal and vertical resolution needed to evaluate the models within the PBL. More observations with higher temporal, spatial and vertical resolution will be an asset for future evaluation of transport models, focusing on intensive campaigns over multiple seasons.***"

**REF-C3:** P5/L6: use longwave instead of long wave
*Author-C3*: Done
"***The members in our multi-physics ensemble all use the same radiation schemes (both longwave and shortwave) but the land surface, surface layer, boundary layer, cumulus, and microphysics schemes are varied for both the inner and the outer domain.***"

**REF-C4:** P12/L22, P13/L5, P15/L21: use the plural form "show" instead of "shows"
*Author-C4*: Done
P12/L22*: "Although the configurations that show the highest RMSE are not always the same across the different variables, these configurations share the same LSM (RUC)."*
P13/L5*: "Similar to the regional RMSE (Figure 5), both LBF and MPX show that the LSM RUC leads to the highest RMSE for the three meteorological variables."*

P15/L21: *"Both wind speed (Figure 13a) and wind direction (Figure 13b) show low correlations, whereas PBLH (Figure 13c) shows consistently positive correlation with the CO₂ mole fraction errors across all sites."*

**REF-C5:** L15/22: "We did not find any relationship between error correlation and distance". It would be convincing to show a scatter plot, error correlation vs. distance for each grid in Figure 13, or at least present the correlation of error correlation vs. distance.
*Author-C5*: Thanks for your suggestion. To address your comment, we decided to follow your suggestion and add a scatter plot with the error correlation vs. distance. We added the following figure (i.e., scatter plot) to the supplement section.

[Figure]

**Figure S1.** Tower and rawinsonde sites specific spatial correlation coefficient between ensemble mean MBE of (a) wind speed, (b) wind direction and (c) PBLH and ensemble mean MBE of DDA CO2 mole fractions versus their distance. The abscissa shows the distance between the rawinsonde and tower sites, while the ordinate shows the spatial correlation.

The following was added to the line cited in the referee comment (see ***bold/italic*** line in the next paragraph):
P15/L19: *"Both wind speed (Figure 13a) and wind direction (Figure 13b) shows low correlations, whereas PBLH (Figure 13c) show consistently positive correlation with the CO₂ mole fraction errors across all sites. **We did not find any relationship between error correlation and distance (see Figure S1).**"*

**REF-C6:** L16/11-12: provide evidence of PBL winds impacting the distribution and magnitude of the inverse CO2 fluxes over the region.
*Author-C6:* In this paper we do not show the impact of PBL wind explicitly on the distribution and magnitude of the CO2 fluxes over the region. However, studies like Lauvaux and Davis (2014) and Deng et al., (2017) show some of this impact. For example, Deng et al. (2017) show the impact that a Four Dimensional Data Assimilation (FDDA) technique has on the different atmospheric variables (e.g., wind speed, temperature) used as input in the atmospheric inversion. Figure 9 from Deng et al. (2017) shows how the influence function vary based on the type of observations assimilated into the model. This figure shows, the influence function when no observations are assimilated (a), only surface stations are assimilated (b), both surface stations and lidar winds are assimilated (c), surface stations, lidar and commercial aircraft data is also assimilated.

[Figure]

*Figure 9. "Influence functions over the INFLUX 1-km resolution domain for 10 of the 12 tower locations of the INFLUX network using the Lagrangian Particle Dispersion Model, averaged for 26–30 October 2013 (corresponding to the observation time 17–22UTC) driven by the meteorological variables from the four different WRF configurations: NOFDDA (Upper left), FDDA_WMO (Upper right), FDDA_WMO_Lidar (Lower left), and FDDA_WMO_Lidar_ACARS (Lower right), in log scale (ppm hour m2 g–1). Note that numbers 1–12 indicate the tower locations as detailed in Figure 8 and two towers were not operational during the time period Oct 26–30. DOI: https://doi.org/10.1525/elementa.133.f9" Note: This caption was copied from Deng et al., (2017) article.*

To this line cited by the referee we added the following citations as an example of this (see ***bold/italic*** line):

P16/L7-L8: "The relationship between PBL winds and $CO_2$ mole fraction is dependent on the local spatial distribution of $CO_2$ surface fluxes and could easily show no clear correlation when averaged over time and space. ***However, we know errors in these two variables can impact the distribution and magnitude of the inverse $CO_2$ fluxes over the region (Deng et al., 2017, Lauvaux and Davis 2014).***"

New citation: Deng, Aijun, Thomas Lauvaux, Kenneth J. Davis, Brian J. Gaudet, Natasha Miles, Scott J. Richardson, Kai Wu et al. "Toward reduced transport errors in a high resolution urban CO 2 inversion system." *Elem Sci Anth* 5 (2017).

**REF-C7:** The reviewer found it difficult to obtain any meaningful information from the figures 7,10&14 where the results of all 45 model results are presented for three selected sites. The results should be first summarized before being presented, or simply be moved to the supplementary.
*Author-C7:* Thanks for the comment, we add a summary of the results prior explaining what we found at each figure. Nest paragraphs shows what we added for the different figures as a summary in ***bold/italic***.

*Author-C7.1:* Figure 7:
P12/L28-L30: "Figure 7 shows the monthly average RMSE of wind speed (Figure 7a-c), wind direction (Figure 7d-f), and PBLH (Figure 7g-i) for each model configuration at specific rawinsonde sites. ***We computed the RMSE for all the different sites (not shown) and we found***

***the highest RMSE in the model configurations that included RUC and Thermal diffusion as the LSM and at some sites when these LSMs are combined with YSU as a PBL scheme.***"

*Author-C7.2:* Figure 10:
P14/L5-L8: "***The MBE analysis was performed for all the sites (not shown), for this statistic we found that all the model configurations show a positive wind speed MBE (overestimation) for the majority of the rawinsonde sites, whereas, wind direction and PBLH shows both positive and negative (underestimation) MBE across the different configuration at the different rawinsonde sites. Some of the positive and negative biases are associated to specific LSMs and PBL schemes.*** Figure 10 show the MBE of three sites that are representative of regional patterns."

*Author-C7.3:* Figure 14
P17/L31-L34: "The sensible heat flux was averaged from 1200 to 2300 UTC and we computed the MBE of the sensible heat flux for the eddy covariance stations close to the rawinsonde sites. ***The MBE was estimated at all the eddy covariance stations available over the region (not shown) and we found that the highest positive sensible heat MBE were found on simulations that used YSU as PBL scheme and RUC or Thermal Diffusion as LSM. Figure 14, shows PBLH MBE of two rawinsonde sites (Figure 14a,b) and the sensible heat MBE of two eddy covariance stations (Figure 14c,d) close to each of these rawinsonde sites.***"